# MobileAIBench: Benchmarking LLMs and LMMs for On-Device Use Cases

## Abstract

The deployment of Large Language Models (LLMs) and Large Multimodal Models (LMMs) on mobile devices has gained significant attention due to the benefits of enhanced privacy, stability, and personalization. However, the hardware constraints of mobile devices necessitate the use of models with fewer parameters and model compression techniques like quantization. Currently, there is limited understanding of quantization's impact on various task performances, including LLM tasks, LMM tasks, and, critically, trust and safety. There is a lack of adequate tools for systematically testing these models on mobile devices. To address these gaps, we introduce MobileAIBench, a comprehensive benchmarking framework for evaluating mobile-optimized LLMs and LMMs. MobileAIBench assesses models across different sizes, quantization levels, and tasks, measuring latency and resource consumption on real devices. Our two-part open-source framework includes a library for running evaluations on desktops and a mobile app for on-device latency and hardware utilization measurements. Our thorough analysis aims to accelerate mobile AI research and deployment by providing insights into the performance and feasibility of deploying LLMs and LMMs on mobile platforms.

## 1 Introduction

With billions of parameters trained on massive amounts of data, LLMs and LMMs have achieved remarkable breakthroughs in a wide range of applications from question answering Baek et al. (2023) to intelligent agents Wang et al. (2024); Liu et al. (2024b) and beyond Achiam et al. (2023); Team et al. (2023); Nijkamp et al. (2022); Murthy et al. (2024). Most recently, the pursuit of deploying LLMs and LMMs on mobile devices has garnered attention, and for good reason. There are several key benefits to deploying AI on mobile devices including offline access, enhanced privacy, and improved performance. It also provides cost efficiency by decreasing server and bandwidth usage, while enhancing user experience with faster and more interactive applications.

Given the extreme limitations of mobile hardware, deploying LLMs and LMMs on mobile devices is challenging. First, the model must have a relatively small number of parameters since parameters drive the number of computations, which consume memory, CPU, and GPU resources. Second, even with a relatively small number of parameters, these models may not fit onto a mobile phone's limited hardware. To further reduce resource footprint, quantization has emerged as a practical, heuristic approach to reduce precision of model weights with seemingly little penalty to performance.

While deploying LLMs and LMMs to mobile devices for real use cases appears to be feasible in the near future, there are knowledge and tooling gaps remaining. First, while quantization seems to be a practical way to reduce the resource footprint of small LLMs and LMMs, there is little to no rigorous measurement and understanding of the effect of quantization on task performance, including LLM tasks, LMM tasks, and critically, trust and safety. Second, there is limited or no tooling available to systematically test quantized models across these tasks. Third, there is limited or no tooling available to test quantized models on a real mobile device across tasks.

In this work, we aim to help accelerate mobile AI research and deployment by providing thorough benchmarking and analysis of open source mobile-optimized LLMs and LMMs. We restrict LLMs and LMMs in consideration to have at most 7B parameters, as we found 7B to be the upper limit of what a high-end phone's hardware can manage (even after quantization). We measure and analyze current LLM and LMM task performance under different levels of quantization, from 16-bit quantization

down to 3-bit quantization in some cases. We selected tasks that are most representative of real-world mobile use cases and considerations. In addition to task performance, we also benchmark our selected LLMs and LMMs on a real mobile device, an iPhone 14. We measure several key latency metrics such as time-to-first-token, and hardware utilization such as CPU usage and RAM usage.

Our results are collected using MobileAIBench, our new two-part framework for evaluating LLMs and LMMs for mobile deployment. The first part of the framework is an open source library, for use on desktops or servers, to evaluate model's performance on a specially selected set of widely known benchmarks. Using this part of the framework, users are able to test their quantized models across benchmarks as they desire. The second component of the MobileAIBench framework is an open-source mobile application, available on both iOS and Android platforms. With our iOS/Android app, users are able to measure latency and mobile hardware utilization such as RAM and CPU of quantized LLMs and LMMs. Our main contributions are summarized as follows:

- We are the first to provide a thorough benchmarking and analysis of open source LLMs and LMMs across varying levels of quantization and various tasks. Our evaluations are generated and reproducible using our newly developed framework, MobileAIBench.

- MobileAIBench is the first open-source framework for on-device task-specific LLM and LMM testing, enabling researchers to evaluate small models and practitioners to assess model viability for mobile deployment.

- We conduct extensive experiments to evaluate LLMs/LMMs over a wide range of tasks, providing insightful findings regarding the impact of quantization and real-mobile deployment.

## 2 RELATED WORK

Many benchmarks have been developed to evaluate LLMs and LMMs from different perspectives. For example, MMLU Hendrycks et al. (2020) provides a large number of tasks to extensively test world knowledge and problem solving ability. AlpacaEval Dubois et al. (2024) and MT-Bench Zheng et al. (2024) provide open-ended question answering evaluation tasks without explicit answers, and employ GPT-4 Achiam et al. (2023) as the success rate judge. KoLA Yu et al. (2023) uses Wikipedia and the continuously collected emerging corpora data to provides knowledge-oriented LLM assessment tasks. TruthfulQA Lin et al. (2021), TrustLLM Sun et al. (2024), Safetybench Zhang et al. (2023) measure LLMs' trust and safety levels. To assist the development of agents, benchmarks Zhou et al. (2023); Chen et al. (2024b); Zhou et al. (2023) have also been developed to evaluate LLM's instruction-following ability. FOFO Xia et al. (2024) contains diverse data formats, and is able to test the format-following ability of current LLMs. MME Fu et al. (2024) provides comprehensive multimodal evaluation benchmars over 14 subtasks. VisIT-Bench Bitton et al. (2023) provides an instruction-following vision-language datasets to test LMMs' real-world use-case. MMVP Tong et al. (2024) provides 9 basic visual patterns that LMMs easily give incorrect answers and hallucinated explanations. However, these benchmarks do not consider the quantized versions of models, nor determine the impact of deployment constraints on model performance. MobileAIBench fills this gap by focusing on deployment utilization on real mobile devices. Besides model performance on specific tasks, MobileAIBench emphasizes model quantization, inference speed, and required deployment resources.

Several papers have discussed strategies and evaluations for developing mobile-ready models Zhang et al. (2024a). Jin et al. (2024) compares different quantization methods and evaluates the performance of quantized LLMs. Liu et al. (2024a) considered different architectures to develop the most performant mobile models. MobileVLM Chu et al. (2023) designs vision language models for mobile devices. The Octopus series Chen et al. (2024a) aim to empower the agentic ability on mobile device by training API tokens. Recent small models such as Phi3 Abdin et al. (2024) and Gemma Team et al. (2024) have, in their model cards, various evaluation results. However, none of these are as comprehensive as the set of data, models, and evaluation metrics proposed in MobileAIBench. In developing this standardized benchmark, we hope to make it easier for model developers to consider the various factors needed to develop mobile-ready models.

## 3 MOBILEAIBENCH FRAMEWORK

MobileAIBench is a two-part evaluation framework for evaluating LLMs and LMMs for mobile deployment. The first part is a pipeline for use on desktops or servers, to evaluate model performance on a specially selected set of widely known benchmarks. The second part is an open source mobile app to measure latency and mobile hardware utilization.

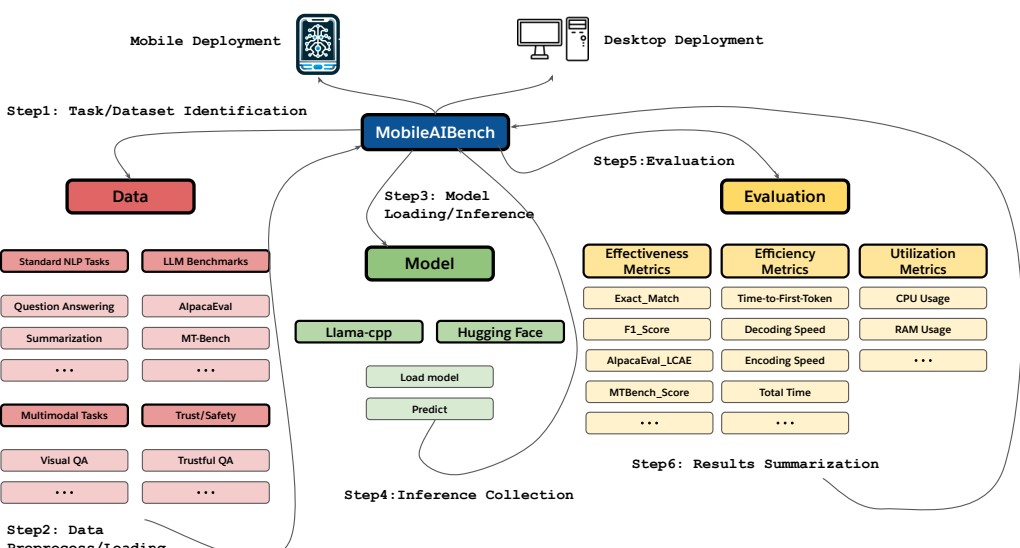

Figure 1: MobileAIBench Architecture

### 3.1 TASKS AND DATASETS

This section outlines the tasks and datasets used to benchmark LLMs & LMMs across various domains including Natural Language Processing (NLP), Multimodal, and Trust & Safety tasks.

#### 3.1.1 STANDARD NLP TASKS

Standard NLP tasks encompass various benchmarks designed to evaluate different capabilities of LLMs. Question answering tests an LLM's ability to comprehend context and respond accurately, for which we use the **Databricks-dolly** Conover et al. (2023) and **HotpotQA** Yang et al. (2018) datasets. Summarization tasks involve condensing large amounts of information into shorter forms while retaining essential ideas. We assess LLM performance in summarization using the **CNN/Daily Mail** Hermann et al. (2015); Nallapati et al. (2016) and **XSum** Narayan et al. (2018) datasets. Text-to-SQL tasks evaluate an LLM's proficiency in crafting SQL queries based on natural language questions. For this purpose, we employ the **Sql-Create-Context** b mc2 (2023) dataset. Additionally, we include popular benchmarks such as the **Massive Multitask Language Understanding (MMLU)** Hendrycks et al. (2021) to evaluate the LLM's accuracy in multitask performance and **Grade School Math (GSM8K)** Cobbe et al. (2021) for assessing LLM's ability to solve mathematical problems. To further evaluate LLM performance, we utilize benchmarks such as **AlpacaEval**, an automatic evaluation method that quantifies the Win-Rate by measuring the proportion of instances where the model's output is preferred over the reference output Li et al. (2023), and **MT-Bench**, a collection of complex multi-turn open-ended questions used to evaluate chat assistants, with GPT-4 serving as the judge Zheng et al. (2023). For Question Answering, Summarization, Text-to-SQL, GSM8K and MMLU tasks, we randomly select 1,000 samples from each relevant dataset. For AlpacaEval and MT-Bench, we use the standard test sets.

### 3.1.2 MULTIMODAL TASKS

Multimodal tasks require LMMs to process different data types such as text, images, and audio. This is critical for developing AI systems that handle complex user requirements on mobile devices. We focus on **Visual Question Answering (VQA)** Antol et al. (2015), selecting five datasets that cover a wide range of contexts. Among them, **VQA-v2** Goyal et al. (2017), **VizWiz** Gurari et al. (2018), **GQA** Hudson & Manning (2019), and **TextVQA** Singh et al. (2019) require the LMM to directly answer visual questions with a single word or phrase. **ScienceQA** Lu et al. (2022) dataset requires selecting the correct answer from multiple choices. Similar to the standard NLP tasks, for each dataset, we randomly select 1000 samples to evaluate the LMMs' performance.

### 3.1.3 TRUST AND SAFETY

To assess the societal impact of LLMs, we include a suite of trust and safety evaluations, focusing on six categories: truthfulness, safety, robustness, fairness, privacy, and ethics, following Sun et al. (2024). For truthfulness, we use the **TruthfulQA** Lin et al. (2021) (TruthQA) dataset to assess the ability to select the correct answer from common misconceptions. For safety, the **Do-Not-Answer** Wang et al. (2023) (DNA) dataset measures if LLMs can refuse illegal, unethical, or otherwise undesirable requests. Robustness is evaluated using **Adversarial Instruction** Sun et al. (2024) (Adv-Inst), which tests models' robustness to prompt perturbations like typos or irrelevant links. Fairness is measured using the **BBQ** dataset Parrish et al. (2021), assessing the tendency to fall for common stereotypes related to gender, age, race, etc. Privacy is tested with hand-crafted prompts based on the **Enron email dataset** Shetty & Adibi (2004) (Priv-Lk), examining if models can decline requests for personal information. For ethics, we use the **Social Chemistry 101** (SC-101) dataset Forbes et al. (2020) to evaluate moral acceptability judgments for different situations.

## 3.2 PART 1: EVALUATION PIPELINE FOR DESKTOP AND CLOUD

The MobileAIBench pipeline, as shown in Figure 1, encompasses three main stages: Data, Model, and Evaluation. In the Data stage, the task and relevant datasets are identified, and the evaluation dataset is created through preprocessing and prompt hydration before being fed into the Model stage. In the Model stage, the model is initialized and predictions are made on the evaluation data, which are subsequently assessed in the Evaluation stage. Various task-specific and generic metrics are supported at the Evaluation stage to gauge the performance of the models. Additionally, MobileAIBench serves as a versatile tool for other researchers and developers to construct their own benchmarking frameworks, thanks to its plug-and-play design, which allows for the easy addition of new tasks and metrics and the creation of custom leaderboards.

**Performance Metrics** are evaluated on desktop for faster inference, with results consistent on mobile devices. These task-oriented metrics assess model effectiveness across various tasks.

## 3.3 PART 2: MOBILE APP

The mobile app component of MobileAIBench is designed to extend the evaluation capabilities to actual mobile devices as shown in Figure 2. This allows for a more accurate assessment of LLM and LMM performance in real-world scenarios. By utilizing the app, we can measure critical efficiency and utilization metrics directly on the device, providing insights into how these models will perform when deployed on end-user mobile hardware. This comprehensive evaluation ensures that the models are not only effective but also efficient and practical for mobile deployment.

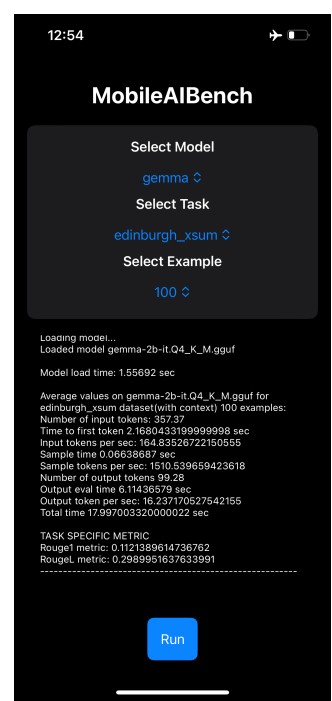

Figure 2: MobileAIBench iOS app

**Mobile Efficiency & Utilization Metrics** are evaluated on real mobile devices using the iOS app to test the deployment feasibility of LLMs and LMMs. **Efficiency** metrics include Time-to-First-Token

(TTFT, s), Input Token Per Second (ITPS, t/s), Output Evaluation Time (OET, t), and Total Time (s). These metrics are model-oriented and measure the efficiency of the models when running on mobile devices. The efficiency is influenced by multiple factors, including model structure, quantization level, and prompt template. **Utilization** metrics measure the resource consumption when running models on real mobile devices. These device-oriented metrics consist of CPU, RAM usage and Battery Drain Rate (BDR, %).

### 3.4 MODEL LIBRARIES SUPPORTED

MobileAIBench is designed to seamlessly test different models on mobile devices. It integrates two inference libraries to accommodate a wide range of LLMs & LMMs: (1) Huggingface[1], which allows users to test any models available on pre-defined tasks by simply changing the model name. Inference with Huggingface provides the performance of the original pre-trained models. (2) Llama.cpp[2], which allows users to test models on real mobile devices. The Llama.cpp inference method supports quantization to reduce model sizes, facilitating deployment on mobile devices.

## 4 BENCHMARKS AND ANALYSIS

For Effectiveness and Trust & Safety experiments, we conducted evaluations on a desktop to assess the impact of various quantization levels. These results are consistent when tested on mobile devices. To evaluate Efficiency and Resource Utilization, we tested the models' performance on an iPhone 14 using our iOS app. The process for selecting evaluation models is detailed in Section A.2.

### 4.1 EFFECTIVENESS EVALUATION

In this section, we examine the effectiveness of various models across different quantization levels (4-bit, 8-bit, and 16-bit). Our primary objective is to evaluate model performance at these bit-width levels without comparing different quantization methods. To ensure consistency in our experimental setup, we used the legacy linear quantization method, commonly known as Q4_0, Q8_0, and f16 in the llama.cpp implementation, as it was available for all the bit-widths under consideration. This approach allowed us to control for variables and focus solely on the impact of quantization levels on model performance. More details on the different quantization methods supported by llama.cpp is presented in Section A.5

#### 4.1.1 STANDARD NLP TASKS

We use several evaluation metrics to assess the performance of the models across different tasks. For question answering tasks, we employ Exact Match (EM) and F1 Score (F1). In the context of Text-to-SQL tasks, we utilize the SQL Parser (SP) and Levenshtein Score (LS). For summarization tasks, we measure performance using Rouge-1 (R1) and Rouge-L (RL). Additionally, we use Win-Rate for AlpacaEval, Score for MT Bench, and Accuracy for both MMLU and GSM8K. Detailed explanations on the implementation of these metrics, along with additional information, are provided in Appendix A.3.2. The performance of various models across different quantization levels is presented in Table 1 and Table 2. In these tables, the highest score for each quantization category is indicated in bold, while the second-best score is underlined. Figure 5 illustrates the violin plots depicting the performance changes when models are quantized from 16-bit to 8-bit. Specifically, Figure 5(a) shows the distribution of performance changes for each model across different tasks, including standard NLP tasks and trust & safety tasks. Figure 5(b) presents the distribution of performance changes for each task when the underlying model shifts from 16-bit to 8-bit.

**Observation and Analysis:** The results indicate that no single model consistently outperforms all others across every task. However, on average, large LLMs (> 6B parameters) exhibit superior performance compared to medium LLMs (1B-6B parameters). While quantization does introduce some performance changes, these changes are not significant in most cases. This finding enhances our confidence in deploying quantized models on mobile devices without substantial performance degradation. In figure 5(a), a narrow distribution and smaller range of performance changes indicate

---

[1]https://huggingface.co/
[2]https://github.com/ggerganov/llama.cpp

Table 1: Effectiveness of LLMs across standard NLP tasks.

| Quantization | Model | Model Size Category | Disk Usage | Question Answering | | | | Text-to-SQL | | Summarization | | | |
| | | | | Databricks | | HotpotQA | | sql-create-context | | CNN | | XSum | |
| | | | | EM | F1 | EM | F1 | SP | LS | R1 | RL | R1 | RL |
|---|---|---|---|---|---|---|---|---|---|---|---|---|---|
| 16bit | Llama 2 7B | > 6B | 13 GB | 0.034 | 0.443 | 0.071 | 0.210 | 0.492 | 0.845 | 0.322 | 0.204 | 0.174 | 0.118 |
| | Mistral 7B | > 6B | 14 GB | 0.043 | **0.498** | **0.137** | **0.267** | 0.485 | 0.770 | 0.328 | 0.204 | 0.170 | 0.114 |
| | Gemma 7B | > 6B | 16 GB | 0.026 | 0.479 | 0.000 | 0.098 | **0.546** | **0.856** | 0.336 | 0.218 | 0.187 | 0.127 |
| | Phi 2 3B | 1B - 6B | 5.2 GB | **0.046** | 0.472 | 0.096 | 0.197 | 0.489 | 0.852 | 0.352 | 0.216 | **0.220** | **0.145** |
| | Gemma 2B | 1B - 6B | 4.7 GB | 0.025 | 0.401 | 0.001 | 0.046 | 0.519 | 0.849 | **0.368** | **0.241** | 0.214 | **0.145** |
| | Zephyr 3B | 1B - 6B | 5.3 GB | 0.032 | 0.441 | 0.030 | 0.109 | 0.457 | 0.787 | 0.325 | 0.203 | 0.169 | 0.112 |
| | TinyLlama 1B | 1B - 6B | 2.1 GB | 0.003 | 0.329 | 0.000 | 0.076 | 0.389 | 0.719 | 0.320 | 0.202 | 0.170 | 0.113 |
| 8bit | Llama 2 7B | > 6B | 6.7 GB | 0.038 | 0.444 | 0.071 | 0.208 | 0.491 | 0.845 | 0.323 | 0.205 | 0.173 | 0.117 |
| | Mistral 7B | > 6B | 7.2 GB | 0.045 | **0.507** | 0.089 | 0.209 | 0.519 | **0.854** | 0.352 | 0.228 | 0.183 | 0.122 |
| | Gemma 7B | > 6B | 8.5 GB | 0.026 | 0.476 | 0.001 | 0.097 | **0.544** | 0.854 | 0.337 | 0.219 | 0.188 | 0.128 |
| | Phi 2 3B | 1B - 6B | 2.8 GB | **0.047** | 0.472 | **0.099** | 0.202 | 0.493 | 0.852 | 0.353 | 0.216 | **0.217** | **0.144** |
| | Gemma 2B | 1B - 6B | 2.5 GB | 0.024 | 0.398 | 0.003 | 0.049 | 0.518 | 0.849 | **0.368** | **0.239** | 0.213 | 0.143 |
| | Zephyr 3B | 1B - 6B | 2.8 GB | 0.032 | 0.440 | 0.030 | 0.108 | 0.449 | 0.782 | 0.324 | 0.203 | 0.170 | 0.113 |
| | TinyLlama 1B | 1B - 6B | 1.1 GB | 0.002 | 0.322 | 0.000 | 0.069 | 0.359 | 0.710 | 0.315 | 0.198 | 0.166 | 0.110 |
| 4bit | Llama 2 7B | > 6B | 3.6 GB | 0.033 | 0.445 | 0.055 | 0.198 | 0.490 | 0.841 | 0.322 | 0.204 | 0.173 | 0.116 |
| | Mistral 7B | > 6B | 3.9 GB | **0.042** | **0.499** | **0.150** | **0.273** | 0.474 | 0.774 | 0.325 | 0.202 | 0.170 | 0.113 |
| | Gemma 7B | > 6B | 4.7 GB | 0.021 | 0.470 | 0.000 | 0.096 | **0.549** | **0.859** | 0.336 | 0.218 | 0.189 | 0.128 |
| | Phi 2 3B | 1B - 6B | 1.5 GB | 0.039 | 0.466 | 0.082 | 0.190 | 0.465 | 0.837 | 0.342 | 0.209 | **0.214** | 0.141 |
| | Gemma 2B | 1B - 6B | 1.5 GB | 0.017 | 0.384 | 0.001 | 0.041 | 0.510 | 0.844 | **0.366** | **0.237** | 0.213 | **0.144** |
| | Zephyr 3B | 1B - 6B | 1.5 GB | 0.035 | 0.445 | 0.027 | 0.106 | 0.459 | 0.789 | 0.330 | 0.206 | 0.170 | 0.112 |
| | TinyLlama 1B | 1B - 6B | 0.6 GB | 0.002 | 0.348 | 0.000 | 0.070 | 0.407 | 0.735 | 0.323 | 0.202 | 0.176 | 0.116 |

a model's robustness to quantization, meaning these models are relatively less sensitive to the quantization process. In contrast, a more spread-out distribution suggests greater sensitivity to quantization, as there is more deviation in performance. The graphs clearly show that different models respond differently to quantization, with some being more sensitive than others. Figure 5(b) demonstrates the robustness of various tasks to the quantization of underlying models, highlighting the importance of considering both model size and sensitivity to quantization when deploying models on edge devices to ensure optimal performance across diverse tasks.

### 4.1.2 MULTI-MODALITY TASKS

We evaluate all LMMs that meet the following criteria: 1) their model structures are supported by the llama.cpp framework (i.e., they have a computational graph implemented based on the GGML library [3]), either by the llama.cpp team or the official model release team, and 2) their model parameter size is smaller than or equal to 7B, thus excluding the models that are unlikely to be deployable on mobile devices in the foreseeable future. Table 2 shows each model's performance on the selected VQA datasets. The evaluation prompt and evaluation metrics for each dataset can be seen in Section A.3.1 and Section A.3.2.

**Observation and Analysis:** Under the original precision (16-bit), no single model outperforms the others across all datasets. On average, Llava-v1.5-7B and BakLLava outperform the others, indicating that larger models have advantages for visual-language understanding. We specifically note that although Moondream2 has only around 1.7B parameters, its performance is highly comparable to the 7B models. It only falls short on SQA, which is the only dataset providing related context besides the image and questions for the models to effectively answer the questions. This may indicate that even the strongest smaller models lack context understanding ability.

We observe that the models' accuracy remains consistent across different quantization levels until 3-bit quantization, where most models experience a significant performance drop, as shown in Figure 3. Moreover, Moondream2 is surprisingly robust to quantization, even at the 3-bit level. This indicates that the effect of quantization on LMMs can vary significantly. Given the importance of quantization for on-device AI, evaluating different models' robustness to quantization is crucial and should be a focus of further study in the AI community.

Disk usage is also an important aspect when deploying models on mobile devices. Therefore, we conducted further evaluation on the trade-off between accuracy and disk usage. As shown in Figure 4, a linear trend indicates that the performance of LLMs generally increases with disk usage, which

---

[3]https://ggml.ai/

is expected as disk usage is highly related to the number of model parameters. The models in the top-left quadrant are considered the best overall regarding both accuracy and size.

Table 2: Effectiveness of LMMs across various VQA datasets.

| Quantization | Model | Model Size | Disk Usage | VQA-v2 | GQA | VisWiz | TextVQA | SQA | Avg. |
|---|---|---|---|---|---|---|---|---|---|
| 16bit | Llava-v1.5-7B | > 6B | 13.13 GB | 0.760 | 0.596 | **0.545** | 0.416 | 0.616 | **0.587** |
| | BakLLaVA | > 6B | 14.07 GB | 0.770 | **0.602** | 0.385 | 0.407 | 0.652 | 0.563 |
| | Llava-phi-2 | 1B - 6B | 5.74 GB | 0.658 | 0.484 | 0.269 | 0.298 | **0.654** | 0.473 |
| | Mobile-VLM-3B | 1B - 6B | 5.63 GB | 0.713 | 0.552 | 0.448 | 0.337 | 0.500 | 0.510 |
| | Mobile-VLM-1.7B | 1B - 6B | 3.12 GB | 0.622 | 0.509 | 0.315 | 0.234 | 0.397 | 0.415 |
| | Moondream2 | 1B - 6B | 3.49 GB | **0.781** | 0.590 | 0.470 | **0.441** | 0.480 | 0.552 |
| 8bit | Llava-v1.5-7B | > 6B | 7.25 GB | 0.759 | **0.600** | **0.545** | 0.419 | 0.606 | **0.586** |
| | BakLLaVA | > 6B | 7.75 GB | 0.770 | 0.597 | 0.384 | 0.402 | 0.655 | 0.562 |
| | Llava-phi-2 | 1B - 6B | 3.31 GB | 0.660 | 0.480 | 0.260 | 0.296 | **0.661** | 0.471 |
| | Mobile-VLM-3B | 1B - 6B | 3.27 GB | 0.714 | 0.566 | 0.460 | 0.335 | 0.490 | 0.513 |
| | Mobile-VLM-1.7B | 1B - 6B | 1.93 GB | 0.619 | 0.515 | 0.308 | 0.241 | 0.401 | 0.417 |
| | Moondream2 | 1B - 6B | 2.26 GB | **0.777** | 0.596 | 0.457 | **0.430** | 0.476 | 0.547 |
| 4bit | Llava-v1.5-7B | > 6B | 4.14 GB | 0.750 | 0.590 | **0.536** | **0.414** | 0.622 | **0.583** |
| | BakLLaVA | > 6B | 4.41 GB | 0.767 | **0.605** | 0.438 | 0.403 | 0.548 | 0.552 |
| | Llava-phi-2 | 1B - 6B | 2.05 GB | 0.658 | 0.475 | 0.247 | 0.281 | **0.646** | 0.461 |
| | Mobile-VLM-3B | 1B - 6B | 2.04 GB | 0.705 | 0.548 | 0.459 | 0.329 | 0.496 | 0.507 |
| | Mobile-VLM-1.7B | 1B - 6B | 1.31 GB | 0.607 | 0.500 | 0.375 | 0.232 | 0.406 | 0.424 |
| | Moondream2 | 1B - 6B | 1.62 GB | **0.780** | 0.587 | 0.446 | 0.411 | 0.479 | 0.541 |
| 3bit | Llava-v1.5-7B | > 6B | 3.33 GB | 0.148 | 0.050 | 0.016 | 0.083 | 0.360 | 0.131 |
| | BakLLaVA | > 6B | 3.53 GB | 0.532 | 0.450 | 0.035 | 0.223 | **0.589** | 0.366 |
| | Llava-phi-2 | 1B - 6B | 1.73 GB | 0.396 | 0.229 | 0.011 | 0.096 | 0.472 | 0.144 |
| | Mobile-VLM-3B | 1B - 6B | 1.71 GB | 0.146 | 0.031 | 0.012 | 0.057 | 0.028 | 0.055 |
| | Mobile-VLM-1.7B | 1B - 6B | 1.15 GB | 0.076 | 0.000 | 0.008 | 0.030 | 0.003 | 0.023 |
| | Moondream2 | 1B - 6B | 1.46 GB | **0.754** | **0.565** | **0.486** | **0.381** | 0.408 | **0.519** |

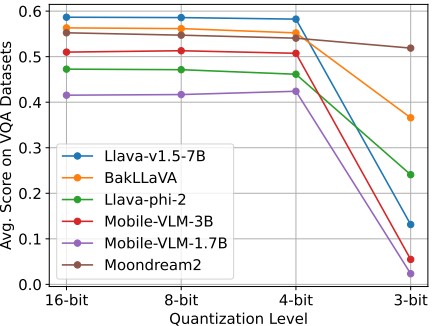

Figure 3: Performance change of LMMs under different quantization.

Figure 4: Trade-off between accuracy and disk usage under 4-bit quantization.

## 4.2 TRUST & SAFETY EVALUATION

We evaluate the performance of various models on trust and safety-related tasks. Ensuring robust performance in these areas is critical, as NLP models are increasingly deployed in sensitive and high-stakes environments. We evaluate the models on various datasets, as discussed in Section 3.1.3. Accuracy is used as the primary effectiveness metric, and we utilize GPT-4o in a llm-as-a-judge framework to determine the accuracy across each of these datasets. The results are shown in Table 3.

**Observation and Analysis:** As with standard NLP tasks, no single model consistently excels across all trust and safety tasks. Larger LLMs (>6B parameters) generally perform better than medium-sized LLMs (1B-6B parameters). While quantization does affect performance, the impact is minimal, indicating that quantized models can be effectively used in trust and safety applications without significant performance degradation.

Table 3: Effectiveness of LLMs across various NLP benchmarks and Trust & Safety datasets.

| Quantization | Model | Model Size Category | Disk Usage | Instruction Task | Multi-turn Chat | Multi-task Eval | Mathematical Reasoning | Trust & Safety | | | | | |
|---|---|---|---|---|---|---|---|---|---|---|---|---|---|
| | | | | AlpacaEval | MT-Bench | MMLU | GSM8K | TruthQA | BBQ | SC-101 | Adv-Inst | DNA | Priv-Lk |
| 16bit | Llama 2 7B | > 6B | 13 GB | 4.918 | 6.259 | 0.418 | 0.271 | 0.272 | 0.341 | 0.626 | **0.943** | **0.855** | **1.000** |
| | Mistral 7B | > 6B | 14 GB | **12.185** | **7.484** | **0.536** | **0.509** | **0.512** | **0.783** | 0.706 | 0.925 | 0.589 | 0.993 |
| | Gemma 7B | > 6B | 16 GB | 1.116 | 4.959 | 0.464 | 0.400 | 0.294 | 0.544 | **0.736** | 0.892 | 0.788 | **1.000** |
| | Phi 2 3B | 1B - 6B | 5.2 GB | 2.932 | 5.318 | 0.479 | 0.190 | 0.370 | 0.588 | 0.676 | 0.849 | 0.277 | 0.020 |
| | Gemma 2B | 1B - 6B | 4.7 GB | 6.459 | 5.187 | 0.352 | 0.135 | 0.187 | 0.310 | 0.594 | 0.916 | 0.830 | **1.000** |
| | Zephyr 3B | 1B - 6B | 5.3 GB | 8.261 | 6.009 | 0.391 | 0.491 | 0.300 | 0.621 | 0.716 | 0.860 | 0.608 | **1.000** |
| | TinyLlama 1B | 1B - 6B | 2.1 GB | 1.529 | 3.962 | 0.168 | 0.110 | 0.153 | 0.236 | 0.628 | 0.879 | 0.366 | 0.530 |
| 8bit | Llama 2 7B | > 6B | 6.7 GB | 5.184 | 5.182 | 0.423 | 0.277 | 0.272 | 0.336 | 0.636 | **0.942** | **0.854** | **1.000** |
| | Mistral 7B | > 6B | 7.2 GB | **8.508** | **6.428** | 0.474 | 0.466 | 0.343 | 0.497 | 0.744 | 0.912 | 0.390 | 0.987 |
| | Gemma 7B | > 6B | 8.5 GB | 1.339 | 4.693 | 0.469 | 0.220 | 0.294 | 0.531 | 0.740 | 0.888 | 0.790 | **1.000** |
| | Phi 2 3B | 1B - 6B | 2.8 GB | 2.972 | 5.740 | **0.482** | 0.229 | **0.367** | 0.580 | 0.674 | 0.847 | 0.274 | 0.27 |
| | Gemma 2B | 1B - 6B | 2.5 GB | 5.713 | 5.328 | 0.352 | 0.146 | 0.192 | 0.307 | 0.592 | 0.910 | 0.831 | **1.000** |
| | Zephyr 3B | 1B - 6B | 2.8 GB | 7.682 | 6.256 | 0.386 | **0.499** | 0.290 | 0.620 | 0.714 | 0.864 | 0.608 | **1.000** |
| | TinyLlama 1B | 1B - 6B | 1.1 GB | 1.514 | 3.562 | 0.188 | 0.160 | 0.158 | 0.256 | 0.612 | 0.846 | 0.389 | 0.367 |
| 4bit | Llama 2 7B | > 6B | 3.6 GB | 4.172 | 6.146 | 0.403 | 0.263 | 0.291 | 0.368 | 0.594 | **0.925** | **0.861** | **1.000** |
| | Mistral 7B | > 6B | 3.9 GB | **13.269** | **7.600** | **0.521** | 0.464 | **0.499** | **0.776** | 0.700 | 0.915 | 0.588 | 0.993 |
| | Gemma 7B | > 6B | 4.7 GB | 1.713 | 4.728 | 0.465 | 0.310 | 0.291 | 0.488 | **0.730** | 0.883 | 0.802 | **1.000** |
| | Phi 2 3B | 1B - 6B | 1.5 GB | 3.583 | 5.750 | 0.477 | 0.224 | 0.371 | 0.613 | 0.702 | 0.817 | 0.309 | 0.087 |
| | Gemma 2B | 1B - 6B | 1.5 GB | 4.835 | 4.946 | 0.346 | 0.117 | 0.181 | 0.289 | 0.588 | 0.914 | 0.817 | **1.000** |
| | Zephyr 3B | 1B - 6B | 1.5 GB | 6.980 | 5.959 | 0.371 | **0.496** | 0.289 | 0.596 | 0.712 | 0.842 | 0.609 | 0.993 |
| | TinyLlama 1B | 1B - 6B | 0.6 GB | 1.320 | 3.534 | 0.167 | 0.140 | 0.135 | 0.252 | 0.636 | 0.859 | 0.363 | 0.080 |

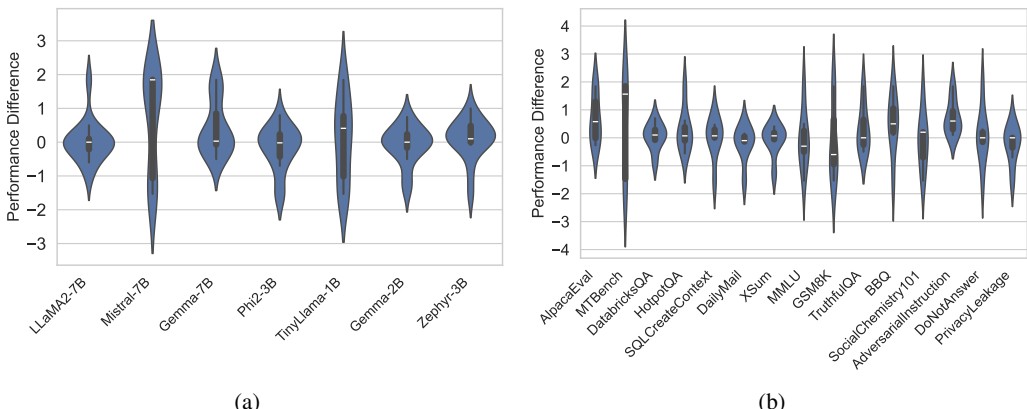

(a)                                         (b)

Figure 5: Distribution of performance changes: (a) per LLM, (b) per task, when transitioning from 16-bit to 8-bit quantization.

### 4.3 EFFICIENCY AND UTILIZATION EVALUATION

We evaluate current LLMs and LMMs on real mobile device using the iOS app provided within MobileAIBench to test the efficiency and utilization. All experiments are evaluated on the same iPhone 14 device to guarantee the comparability. All models are quantized to 4bit and only those under 3B are deployable. We conduct the experiments with 3 LLMs (Phi2 3B, Gemma 2B, TinyLlama 1B) on 4 NLP datasets and 1 LMM (Llava-Phi-2) on 2 LMM datasets.

**Observation and Analysis:** Experiment results of LLMs are shown in Table 4. We can have the following observations. (1) Smaller models have lower TTFT and higher ITPS/OTPS. It indicates smaller models have faster encoding/decoding speed as the computation required for processing each token is decreased. (2) The shortest OET and Total Time may not be achieved by the smallest model. For example, on HotpotQA dataset, the lowest OET and Total Time are achieved by Phi2 model. It owes to the input/output token length that is directly related with the model. Though Phi2 has the slowest encoding/decoding speed, it can achieve the fastest OET and Total time by giving more concise responses. (3) The on-device memory consumption is intense and directly relates to model sizes. For the iPhone 14 with a total RAM of 6 GiB, even running the 4-bit quantized TinyLlama model (1B) on-device takes more than 50% of the overall memory, leaving limited space for other APPs. The memory consumption increases with larger models. (4) The CPU utilization is naturally different with different models. On all the 4 datasets, Phi2 takes the lowest and Gemma takes the highest CPU utilization, respectively. It is a surprise finding that the on-device CPU utilization is not related to model sizes, which reals the necessity of on-device testing for mobile deployment.

Table 4: Efficiency & Utilization of LLMs across NLP tasks. Metrics explanation in Section 3.3

| Dataset | Model | Efficiency | | | | | Utilization | | |
|---|---|---|---|---|---|---|---|---|---|
| | | TTFT(s) | ITPS(t/s) | OET(s) | OTPS(t/s) | Total Time(t) | CPU(%) | RAM(GiB) | BDR(%) |
| HotpotQA | Phi2 3B | 2.32 | 94.04 | **1.78** | 13.21 | **5.02** | **63.89** | 4.33 | 9.33 |
| | Gemma 2B | 2.86 | 133.35 | 3.52 | 13.65 | 11.62 | 85.3 | 4.25 | 10.22 |
| | TinyLlama 1B | **1.60** | **277.24** | 3.12 | **28.14** | 6.30 | 71.58 | **3.34** | **5.37** |
| Databricks-dolly | Phi2 3B | 2.01 | 88.39 | 3.35 | 12.74 | 7.01 | **61.28** | 4.17 | 11.76 |
| | Gemma 2B | 1.93 | 168.49 | **2.83** | 16.31 | 9.25 | 87.66 | 4.24 | 18.51 |
| | TinyLlama 1B | **1.08** | **345.32** | 2.88 | **31.24** | **5.45** | 70.47 | **3.34** | **10.00** |
| Sql-create-context | Phi2 3B | 0.73 | 96.53 | **1.72** | 13.84 | **3.36** | 76.67 | 4.37 | 9.09 |
| | Gemma 2B | 0.75 | 163.01 | 2.14 | 16.77 | 6.56 | 99.17 | 4.45 | 21.6 |
| | TinyLlama 1B | **0.39** | **349.05** | 2.08 | **32.89** | 3.61 | 80.31 | **3.37** | **7.50** |
| XSum | Phi2 3B | 2.73 | 73.40 | 8.41 | 11.66 | 15.13 | **68.35** | 4.57 | 18.66 |
| | Gemma 2B | 2.30 | 154.88 | 6.37 | 15.56 | 18.99 | 94.00 | 4.44 | 36.17 |
| | TinyLlama 1B | **1.29** | **321.70** | **4.10** | **30.13** | **7.45** | 70.88 | **3.43** | **15.00** |

(5) Battery Drain Rate (BDR) increases with both model size and the number of output tokens generated. Larger models and longer outputs consume more battery power, highlighting the need for optimization in model design and output length to improve energy efficiency.

The results for LMMs are presented in Table 5. Since the number of output tokens is only one for both datasets, we do not have OET and OPTS efficiency results, and the TTFT is equivalent to the Total Time. As observed, multimodal tasks are significantly more computation-intensive compared to NLP tasks. The average TTFT ex-

Table 5: Efficiency & Utilization of LMMs.

| Dataset | Model | Samples | Efficiency | | Utilization | |
|---|---|---|---|---|---|---|
| | | | TTFT(s) | ITPS(t/s) | CPU(%) | RAM(GiB) |
| VQA-v2 | Llava-Phi-2 | 10 | **66.47** | **1.24** | **51.06** | **4.63** |
| | | 25 | 213.48 | 0.37 | 51.29 | 4.66 |
| | | 50 | 350.06 | 0.22 | 57.62 | 4.67 |
| ScienceQA | Llava-Phi-2 | 10 | **81.00** | **4.65** | 90.39 | **4.56** |
| | | 25 | 223.61 | 1.55 | 78.46 | 4.63 |
| | | 50 | 508.66 | 0.68 | **77.52** | 4.65 |

ceeds 60 seconds for both tasks, and the ITPS is lower than 5. This indicates that current LMMs may not yet be suitable for mobile deployment. However, this could improve with newer mobile devices with enhanced computational capabilities. Additionally, we notice a decrease in efficiency with an increasing number of samples, likely due to the elevated temperature resulting from processing more samples. In Section A.4.2, we conduct a comprehensive latency analysis of the LMMs on Intel CPUs to further compare their latency with each other. The RAM usage keeps nearly constant after loading the model, which also matches the observation of NLP tasks in Section A.4.1.

## 5 DISCUSSION

MobileAIBench offers a comprehensive evaluation of LLMs and LMMs in terms of task effectiveness, on-device efficiency, and utilization. It covers a wide range of text and multimodal tasks, assessing the impact of various quantization levels on current models. Additionally, trust and safety evaluations are included to ensure the reliability and security of LLM/LMM applications on mobile devices. As a benchmarking platform for mobile deployment, MobileAIBench aims to enhance performance while minimizing any potential negative social impact.

Extensive experiments with MobileAIBench reveal several interesting findings. Quantization is an effective way to decrease model size while keeping the performance. Different models/tasks have varied sensitivity to quantization levels. Current LLMs and LMMs require significant resources in terms of CPU and RAM usage when deployed on mobile devices, even with the 1B model. More compact and optimized LLMs and LMMs are needed for mobile deployment.

While MobileAIBench is a significant advancement, we acknowledge its limitations. It excludes other model compression methods, like model pruning, as they are not mature enough for deployment. Additionally, due to space constraints and the wide variety of mobile devices, we have not conducted a comprehensive comparison across different hardware. Support for different species of mobile devices is still under development and will open to public upon release.

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

# A  APPENDIX

## A.1  DATASET

Datasets within MobileAIBench are summarized in Table 6. We select tasks with real-world Mobile use cases. These 20 datasets cover NLP, Multi-modality and Trust&Safety tasks, providing a comprehensive evaluation for mobile use cases. To support runable experiments on mobile devices, we down-sample the origional datasets to mostly 1000 samples.

Table 6: Summary of datasets integrated within MobileAIBench.

| Task-type | Tasks | Datasets | # Samples | Avg. # Length |
|---|---|---|---|---|
| NLP | Question Answering | Databricks-dolly Conover et al. (2023) | 1000 | 156.90 |
| | | HotpotQA Yang et al. (2018) | 1000 | 443.59 |
| | Summarization | CNN/Daily MailHermann et al. (2015); Nallapati et al. (2016) | 1000 | 482.03 |
| | | XSum Narayan et al. (2018) | 1000 | 298.69 |
| | Text-to-SQL | Sql-create-context b mc2 (2023) | 1000 | 18.61 |
| | Language Understanding | MMLU Hendrycks et al. (2020) | 1000 | 94.82 |
| | Math | GSM8K Cobbe et al. (2021) | 1000 | 72.35 |
| | LLM Benchmarks | Alpacaeval Li et al. (2023) | 805 | 28.56 |
| | | MTBench Zheng et al. (2024) | 80 | 66.97 |
| Multi-modality | Direct Answer VQA | VQA-v2 Goyal et al. (2017) | 1000 | 15.18 |
| | | VizWiz Gurari et al. (2018) | 1000 | 25.20 |
| | | GQA Hudson & Manning (2019) | 1000 | 17.55 |
| | | TextVQA Singh et al. (2019) | 1000 | 16.05 |
| | Multiple-Choice VQA | ScienceQA Lu et al. (2022) | 1000 | 59.95 |
| Trust & Safety | Truthfulness | TruthfulQA Lin et al. (2021) | 817 | 71.23 |
| | Safety | Do-Not-Answer Wang et al. (2023) | 1000 | 14.33 |
| | Robustness | Adversarial Instruction Sun et al. (2024) | 600 | 9.70 |
| | Fairness | BBQ Parrish et al. (2021) | 1000 | 67.35 |
| | Privacy | Privacy Leakage Shetty & Adibi (2004) | 150 | 11.25 |
| | Ethics | Social Chemistry 101 Forbes et al. (2020) | 500 | 22.05 |

## A.2  EXPERIMENT MODEL SELECTION

**LLM Model Selection:** We select several LLMs of varied sizes and architectures to evaluate their performance and utilization on mobile devices. The selected models are categorized into large-size (>6B) and medium-size (1B-6B) groups. For large-size LLMs, we include Llama2 Touvron et al. (2023), Mixtral-7B Jiang et al. (2023), and Gemma-7B Team et al. (2024). Among medium-sized LLMs, we choose Phi-2 Javaheripi et al. (2023), TinyLlama-1.1B Zhang et al. (2024b), Gemma-2B Team et al. (2024), and StableLM-3B stability.ai (2024).

**LMM Model Selection:** We categorize the selected models into two groups: Large Models (> 6B parameters), including Llava-v1.5-7B and BakLLava-7B, and Medium Models (1B-6B parameters), which consist of Llava-phi-2, Mobile-VLM-3B, Mobile-VLM-1.7B, and Moondream2.

The LLMs considered for mobile testing include Phi-2, Gemma-2B, and TinyLlama-1.1B, all quantized to 4-bit to ensure compatibility with the iPhone 14's resource constraints.

We note that serving LMMs on mobile devices is significantly more challenging than serving text-based LLMs due to their ensemble structure, larger model sizes, and complex inference processes. Therefore, we are not limiting the model selection to those fully supported by Llama.cpp, at this time. Instead, we select small-sized LMMs that are likely to be supported in the foreseeable future.

## A.3  EVALUATION DETAILS

Various evaluation metrics are considered for Standard NLP tasks, Multi-modality, and Trust & Safety. The details of the same are provided below.

### A.3.1 EVALUATION PROMPT TEMPLATES

The evaluation prompts for various NLP datasets are listed in Table 7, while those for various multimodal datasets are listed in Table 8.

### A.3.2 EVALUATION METRICS

- **Task: Question & Answering**
  - **Exact Match:** The Exact Match score evaluates the accuracy of a QA system by checking whether the predicted answer is exactly the same as the reference answer. This means that for each question, the predicted answer must match the ground truth answer exactly, including any formatting, punctuation, and whitespace.
  - **F1 Score:** F1 score combines precision and recall into a single metric by taking their harmonic mean. We tokenize the ground truth and predictions, then we computer the precision and recall and finall compute the F1 score.

- **Task: Summarization**
  - **Rouge-1:** The ROUGE-1 metric measures the overlap of unigrams (single words) between the candidate summary and the reference summary.
  - **Rouge-L:** The ROUGE-L metric evaluates the longest common subsequence (LCS) between the candidate summary and the reference summary. It measures the precision, recall, and F1 score based on the longest matching sequence of words, emphasizing the importance of order and continuity in the generated summaries.

- **Task: Text-to-SQL**
  - **SQL Parser:** The SQL query is converted into a graph where each node represents an SQL keyword, and its children represent the items associated with that keyword. This is implemented using a Python dictionary. To measure the SQL parser score, we evaluate the overlap between the ground truth graph and the predicted graph.
  - **Levenshtein score:** The Levenshtein score, also known as the Levenshtein distance, measures the minimum number of single-character edits (insertions, deletions, or substitutions) required to change one word or string into another.

- **Task: AlpacaEval**
  - **Win-Rate:** Fraction of times predicted response is chosen over the predictions made by a baseline model. (In our case, baseline model is GPT-4)

- **Task: MT-Bench**
  - **Score:** Using GPT-4 in llm-as-a-judge framework, we ask the judge to score a given response on a scale of 10. (10 being the highest score)

- **Task: Trust & Safety**
  - **Accuracy:** Using GPT-4o in llm-as-a-judge framework, we ask the judge to determine if the predicted answer is same as the ground truth answer.

- **Task: VQA**
  - **Score (single ground-truth):** For datasets with a single ground-truth answer per question (e.g., GQA, ScienceQA), we score each test sample based on exact match: $1$ if the prediction matches, otherwise $0$. Then, the score for each dataset is calculated by averaging the total test cases.
  - **Score (multiple ground-truth):** For datasets with multiple human-provided ground-truth answers (e.g., VQA-v2, VisWiz, TextVQA), the accuracy score for each test case is calculated as: $\min\left(\frac{\text{number of matches}}{3}, 1\right)$. This follows the official evaluation design of VQA-v2.

## A.4 EXPERIMENT RESULTS

### A.4.1 UTILIZATION EXPERIMENTS

When running experiments on-device with MobileAIBench, we also obtain the trace files for CPU/Memory utilization. LLMs experiment trace files are shown in Figure 6. From the utilization

Table 7: Evaluation prompts for different NLP datasets.

| Dataset | Evaluation Prompt |
|---|---|
| Llama 2 7B | `[INST] «SYS» {system} «/SYS» {prompt} [/INST]` |
| Mistral 7B | `[INST] {system} {prompt} [/INST]` |
| Gemma 7B | `<start_of_turn>user\n{system} {prompt}<end_of_turn>\n<start_of_turn>mode` |
| Zephyr 3B | `<|user|>\n{system}\n{prompt}<|endoftext|>\n<|assistant|>\n` |
| Phi2 3B | `{system} Instruct:{prompt}\nOutput` |
| Gemma 2B | `<start_of_turn>user\n{system} {prompt}<end_of_turn>\n<start_of_turn>mode` |
| TinyLlama 1B | `<|system|>\n{system}\n<|user|>\n{prompt}\n<|assistant|>` |

Table 8: Evaluation prompts for different VQA datasets.

| Dataset | Evaluation Prompt |
|---|---|
| VQAV2 | `[Question]`\n Answer the question using a single word or phrase. |
| VisWiz | `[Question]`\n When the provided information is insufficient, respond with 'Unanswerable'. Answer the question using a single word or phrase. |
| GQA | `[Question]`\n Answer the question using a single word or phrase. |
| TextVQA | `[Question]`\n Answer the question using a single word or phrase. |
| ScienceQA | Context: `[context]`\n Question:`[question]`\n Options: (A)`[Option Content]` (B) `[Option Content]`...\n Answer with the option letter from the given choices directly. |

trace file, we can observe the changes when running LLMs/LMMs on device. We can observe that on the four datasets, CPU utilization is not stable, and has a large fluctuations on the utilization curve. That is caused by the different CPU utilization when loading/inferencing samples. However, the memory keeps nearly constant during inference, which shows models take the majority of memory and the memory cost for samples are relatively limited. Same trace files can also be obtained for LMM tasks, and we omit them for similar observations.

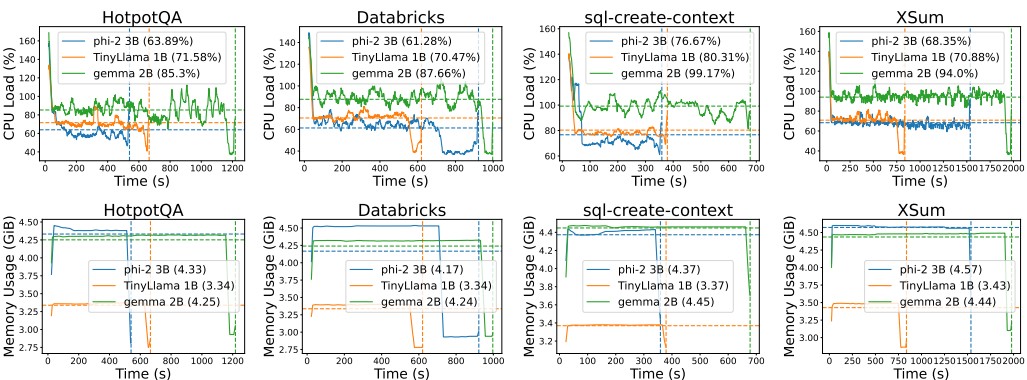

Figure 6: CPU/Memory trace of different LLMs.

### A.4.2 LATENCY ANALYSIS FOR LMMS ON INTEL CPU

Given the computational limitations of current mobile devices, it is challenging to conduct comprehensive on-device latency evaluations for all LMMs. To compare the latency of these models and provide information about their latency-related performance, we conduct latency evaluations on Intel CPUs of model type Intel(R) Xeon(R) CPU @ 2.20GHz. The results are shown in Table 9.

The results indicate that quantizing the models to 8-bit and 4-bit levels generally improves inference speed. However, 3-bit quantization does not result in faster performance compared to the original model.

Table 9: Latency (time to first token) comparison of LMMs with different quantization levels.

| Model | 16bit | 8bit | 4bit | 3bit |
|---|---|---|---|---|
| Llava-v1.5-7B | 12.620 | 9.886 | 11.345 | 12.683 |
| BakLLaVA | 14.099 | 10.515 | 11.986 | 13.851 |
| Llava-phi-2 | 6.513 | 5.398 | 6.025 | 7.030 |
| Mobile-VLM-3B | 2.917 | 2.534 | 2.690 | 2.956 |
| Mobile-VLM-1.7B | **2.003** | **1.889** | **1.901** | **2.054** |
| Moondream2 | 4.008 | 3.819 | 3.914 | 5.217 |

Among all the models, smaller ones generally exhibit lower latency. Additionally, the results clearly demonstrate that the Mobile-VLM series has significantly lower latency compared to other models of similar size. According to the original paper Chu et al. (2023), the low latency of Mobile-VLM is attributed to the application of an additional convolutional layer after the visual tokens, which reduces the number of image tokens by a factor of four. This approach could be a viable strategy for developing latency-driven LMMs.

A.5   VARIOUS QUANTIZATIONS SUPPORTED BY LLAMA.CPP

The llama.cpp library supports a wide variety of quantization levels for efficient model compression and inference. These include:

- **Q4_0 and Q4_1**: 4-bit quantization, which provides a good balance between performance and model size.

- **Q5_0 and Q5_1**: 5-bit quantization, offering slightly higher precision than Q4.

- **Q8_0**: 8-bit quantization for scenarios where maintaining higher accuracy is crucial.

- **Q2_K, Q3_K, Q4_K, Q5_K, and Q6_K**: Variants of k-quantizations, which are more specialized, aiming to reduce size further with minimal impact on accuracy.

- **F16 and BF16**: Floating-point formats for higher precision without the full overhead of 32-bit float types.

- **F32**: The highest precision format, which is typically the baseline.

```
Allowed quantization types:
   2  or  Q4_0   :  3.50G, +0.2499 ppl @ 7B - small, very high quality loss - legacy, prefer using Q3_K_M
   3  or  Q4_1   :  3.90G, +0.1846 ppl @ 7B - small, substantial quality loss - legacy, prefer using Q3_K_L
   8  or  Q5_0   :  4.30G, +0.0796 ppl @ 7B - medium, balanced quality - legacy, prefer using Q4_K_M
   9  or  Q5_1   :  4.70G, +0.0415 ppl @ 7B - medium, low quality loss - legacy, prefer using Q5_K_M
  10  or  Q2_K   :  2.67G, +0.8698 ppl @ 7B - smallest, extreme quality loss - not recommended
  12  or  Q3_K   :  alias for Q3_K_M
  11  or  Q3_K_S :  2.75G, +0.5505 ppl @ 7B - very small, very high quality loss
  12  or  Q3_K_M :  3.06G, +0.2437 ppl @ 7B - very small, very high quality loss
  13  or  Q3_K_L :  3.35G, +0.1803 ppl @ 7B - small, substantial quality loss
  15  or  Q4_K   :  alias for Q4_K_M
  14  or  Q4_K_S :  3.56G, +0.1149 ppl @ 7B - small, significant quality loss
  15  or  Q4_K_M :  3.80G, +0.0535 ppl @ 7B - medium, balanced quality - *recommended*
  17  or  Q5_K   :  alias for Q5_K_M
  16  or  Q5_K_S :  4.33G, +0.0353 ppl @ 7B - large, low quality loss - *recommended*
  17  or  Q5_K_M :  4.45G, +0.0142 ppl @ 7B - large, very low quality loss - *recommended*
  18  or  Q6_K   :  5.15G, +0.0044 ppl @ 7B - very large, extremely low quality loss
   7  or  Q8_0   :  6.70G, +0.0004 ppl @ 7B - very large, extremely low quality loss - not recommended
   1  or  F16    : 13.00G              @ 7B - extremely large, virtually no quality loss - not recommended
   0  or  F32    : 26.00G              @ 7B - absolutely huge, lossless - not recommended
```

Figure 7: Quantizations supported by llama.cpp

We have used the legacy linear quantization method, commonly referred to as Q4_0, Q8_0, and f16 in the llama.cpp implementation for benchmarking on desktop and cloud. We selected this legacy method for benchmarking purposes because it was available for all the bit-widths under consideration (4-bit, 8-bit, and 16-bit), ensuring consistency in our experimental setup. Using a single quantization

technique across different bit-widths allowed us to control for variables and focus solely on the impact of quantization levels on model performance.

## A.6 MOBILE APP DEVELOPMENT

For on-device development using llama.cpp, the recommended quantization levels are primarily Q4_M, Q5_K_S, and Q5_K_M. These levels provide a good balance between model size and output quality while minimizing perplexity loss. We have used Q4_K_M method for on-device mobile app development, due to its balanced trade-off between model size and output quality. It offers a reduced memory footprint and faster inference speeds while maintaining relatively low quality loss.

We have also released an android version of the same app, screenshot below. This will allow users to compare model performance across a broader range of hardware configurations, enhancing the utility of our framework.

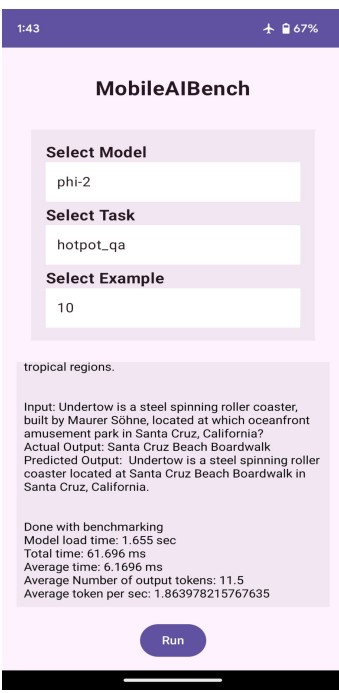

Figure 8: Screenshot of MobileAIBench Android app

