# OpenReview forum: "MobileAIBench: Benchmarking LLMs and LMMs for On-Device Use Cases"
_ICLR.cc/2025/Conference — Submitted to ICLR 2025_

### Official Review · Reviewer_NJfu · 2024-11-03

**Soundness:** 3
**Presentation:** 3
**Contribution:** 2
**Rating:** 6
**Confidence:** 4

**Summary:**

This paper proposes MobileAIBench, a platform for evaluating the performance of large language models (LLMs) and multimodal models (LMMs) on mobile devices. MobileAIBench focuses on the task performance, resource consumption, and trust and security of quantized models on mobile devices. The authors have built testing tools for desktop and mobile platforms and explored the impacts of different quantization levels on task effectiveness and resource utilization through experiments. The platform has reference value in the mobile AI application scenarios.

**Strengths:**

**Comprehensive mobile AI testing platform**:  MobileAIBench integrates the existing benchmark testing framework for tasks and models. It is suitable for performance evaluation on end devices and fills the gap in mobile large model evaluation.

**Multi-dimensional performance evaluation**: The platform not only tests the performance of models on standard NLP and multimodal tasks, but also covers trust and security dimensions, highlighting privacy protection, bias, and ethical issues in mobile deployment.

**Real-device testing**:  MobileAIBench tests LLMs directly on mobile devices with key metrics such as latency, memory usage, CPU utilization, and battery consumption, which makes the results closer to actual application scenarios.

**Weaknesses:**

**Insufficient mobile-side experiments**:  The focus of MobileAIBench should be on deploying LLMs and LMMs on mobile devices and examining the effects of quantization on task performance in these environments. However, most of the experiment results are from desktops rather than mobile devices. Supplementing with more mobile-side experiments, such as assessing the impacts of quantization strategies on mobile devices, would strengthen the work.

**Lack of data-level innovation**: MobileAIBench seems to be a collection of existing tasks and datasets without introducing specific data or design for benchmarking LLMs in mobile scenarios. As a benchmark, it lacks specialized datasets or test case designs tailored to mobile-specific scenarios, which would better demonstrate the platform’s value. Thus, it may be more suitable to position MobileAIBench as a platform or testing tool rather than a standalone benchmark.

**Claims of device consistency require support**:  In line 235, the authors assert that test results on desktop devices are consistent with those on mobile devices. However, the paper does not provide sufficient experimental data to support this claim.

**Room for improvement in experimental design**:  Although the potential impacts of device temperature is mentioned (line 462), temperature should be included as a sub-metric in the evaluation metrics to better reflect real-world conditions on mobile devices. I still recommend to test on more mobile devices to obtain more persuasive results.

**Questions:**

Can the authors provide more discussions and justifications for the above?

---

> ### Author Response · Authors · 2024-11-26
>
> ## Weaknesses / Questions:
> > Insufficient mobile-side experiments: The focus of MobileAIBench should be on deploying LLMs and LMMs on mobile devices and examining the effects of quantization on task performance in these environments. However, most of the experiment results are from desktops rather than mobile devices. Supplementing with more mobile-side experiments, such as assessing the impacts of quantization strategies on mobile devices, would strengthen the work.
>
> We appreciate the reviewer's feedback regarding the need for more mobile-side experiments. Our current work includes mobile-specific evaluations leveraging an iOS app to directly assess key efficiency and utilization metrics such as Time-to-First-Token (TTFT), Input Tokens Per Second (ITPS), CPU/RAM usage, and Battery Drain Rate (BDR) on real mobile devices. These experiments evaluate quantized models across various tasks, including standard NLP (HotpotQA, Databricks-Dolly, Sql-Create-Context, XSum) and multimodal tasks (VQA-v2, ScienceQA), providing an accurate understanding of model performance in mobile environments (Section 4.3, Tables 4 and 5). However, there were several important technical constraints that shaped our experimental design. Currently, only models under 3B parameters can be practically deployed on mobile devices even after quantization, due to the significant memory and computational limitations of current mobile hardware. As shown in our experiments on the iPhone 14, even running a 4-bit quantized TinyLlama model (1B parameters) consumes over 50% of the device's 6GB RAM.
>
> Given these hardware constraints, we took a two-pronged approach: First, we conducted comprehensive quantization impact analysis on desktop/cloud to establish baseline performance impacts across model sizes and architectures. This allowed us to evaluate larger models (up to 7B parameters) and identify promising candidates for mobile deployment. Then, for models that could run on mobile devices (Phi2, Gemma 2B, and TinyLlama 1B), we performed detailed on-device experiments measuring critical metrics like time-to-first-token, CPU/RAM utilization, and battery drain across multiple tasks.This approach lets us provide both broad insights about quantization effects and specific mobile deployment metrics.
>
> > Lack of data-level innovation: MobileAIBench seems to be a collection of existing tasks and datasets without introducing specific data or design for benchmarking LLMs in mobile scenarios. As a benchmark, it lacks specialized datasets or test case designs tailored to mobile-specific scenarios, which would better demonstrate the platform’s value. Thus, it may be more suitable to position MobileAIBench as a platform or testing tool rather than a standalone benchmark.
>
> While it's true that we leverage existing datasets, this was an intentional design choice to ensure comparability with established benchmarks while adding crucial mobile-specific evaluation dimensions. Our framework's innovation lies not in creating new datasets, but in providing the first comprehensive tooling and methodology for evaluating LLMs and LMMs specifically for mobile deployment - a critical gap in current research infrastructure. The selection of existing datasets was carefully curated to represent real-world mobile use cases, spanning question-answering, summarization, visual understanding, and trust & safety evaluation. Using established datasets allows researchers to contextualize mobile performance against known baselines while our framework adds critical new mobile-specific metrics such as time-to-first-token, CPU/RAM utilization, and battery drain that are absent from existing benchmarks. However, we acknowledge the reviewer's point about mobile-specific scenarios and agree this represents an opportunity for future work.

---

> > ### Author Response · Authors · 2024-11-26
> >
> > > Claims of device consistency require support: In line 235, the authors assert that test results on desktop devices are consistent with those on mobile devices. However, the paper does not provide sufficient experimental data to support this claim.
> >
> > Thank you for highlighting this important point about our consistency claim. The assertion in line 235 about consistency between desktop and mobile device results was based on preliminary observations during our experiments, specifically regarding the relative performance rankings and comparative trends between models under different quantization levels. However, we acknowledge that this claim requires more robust supporting evidence and detailed experimental validation. Currently, we can only verify this consistency for models under 3B parameters that can actually run on mobile devices (like Phi2, Gemma 2B, and TinyLlama 1B), as shown in Tables 4 and 5.
> >
> > > Room for improvement in experimental design: Although the potential impacts of device temperature is mentioned (line 462), temperature should be included as a sub-metric in the evaluation metrics to better reflect real-world conditions on mobile devices. I still recommend to test on more mobile devices to obtain more persuasive results.
> >
> > We appreciate the reviewer's valuable feedback regarding temperature considerations and device diversity in our experimental design. We agree that device temperature is a critical factor in mobile AI deployment that deserves more thorough investigation. While we observed temperature impacts on efficiency metrics during our experiments (particularly noticeable in the decreased performance with increasing sample numbers for LMMs, as noted in Section 4.3), we acknowledge that a more systematic approach to temperature measurement and analysis would strengthen our evaluation framework. We plan to enhance MobileAIBench by incorporating temperature as a formal sub-metric, including continuous temperature monitoring during model inference, analysis of thermal throttling impacts on performance, and examination of the relationship between model size, quantization level, and heat generation. Regarding device diversity, our current results from the iPhone 14 provide important initial insights, but we agree that testing across a broader range of devices with varying hardware capabilities and thermal characteristics would provide more comprehensive and generalizable results. In future work, we plan to expand our evaluation to include different iPhone models, Android devices across various price points and hardware configurations, and tablets, which would help establish more robust benchmarking standards for mobile AI deployment. This expanded device coverage would also help identify how different mobile hardware architectures and thermal management systems impact model performance.

---

> > > ### Comment · Reviewer_NJfu · 2024-12-03
> > >
> > > Thank the authors for providing the detailed responses to the concerns and questions. After reading the others' comments as well, I would lean to rate more positive.

---

### Official Review · Reviewer_1eAF · 2024-11-04

**Soundness:** 3
**Presentation:** 3
**Contribution:** 4
**Rating:** 6
**Confidence:** 3

**Summary:**

A new benchmark is introduced to evaluate the behavior of LLMs and LMMs across various quantization levels, simulating deployment scenarios on mobile devices. By utilizing standard NLP, VQA, and safety tasks, the authors provide benchmarking references that offer insights into how model performance varies with quantization. The study highlights both the effectiveness and limitations of current quantization methods for LLMs. The limitation includes, for example, the need to address efficiency challenges in LMMs. The experimental findings underscore differences in performance across quantization levels, providing valuable information for developing more efficient algorithms.

The key contributions of this work are a novel benchmark regarding the quantization of LLM/LMMs, an open-source platform running on real devices, and in-depth analyses of the current quantization method and models.

**Strengths:**

S1 - This work makes a valuable contribution by expanding the community's understanding of models’ behaviors with quantization. This offers several analyses that will be beneficial for further research in this area, especially when deploying the model on mobile devices.

S2 - The open-sourced experimental platform is highly meaningful, allowing others to reproduce the work easily.

S3 - For analyses, the authors have taken a comprehensive approach by considering various evaluation axes. For example, the studies on multimodal and safety tasks enhance the study’s relevance and depth.

**Weaknesses:**

W1 - The current set of tasks can be limited. With the growing interest in UI-based control for digital devices (such as Cluade-3.5 for computer use), it would be beneficial to include related tasks. Have the authors considered incorporating AndroidWorld (Rawles et al., 2024) for general capability assessment or MobileSafetyBench (Lee et al., 2024) for evaluating the safety of agents controlling mobile devices?

W2 - Relying solely on VQA for multimodal tasks may restrict the scope of analysis. Including other tasks, such as image captioning or OCR, could provide a more comprehensive evaluation of capabilities, especially considering their usage on mobile devices.

W3 - Although the authors’ choice of the iPhone-14 as a representative device is understandable, it would enhance the robustness of the study to consider other device types. For example, assessment with Android OS devices or tablets would provide a broader understanding.

W4 - (Minor) Certain aspects of the presentation could be improved. For example, the explanation of Figure 5 could be more detailed, and Figure 7 appears to be oddly rendered.

**Questions:**

Q1 - Could the authors clarify how this benchmark compares with existing ones regarding LLM quantization, such as LLM-QBench (Gong et al., 2024)? This comparison would help readers understand the unique contributions and positioning of this benchmark.

Q2 - What was the rationale behind selecting a random sample of 1,000 from the dataset? Justification of this choice, particularly regarding the representativeness and generalizability of the results, would be valuable.

Q3 - Could the authors explain why Llama-3.1-8B was not included in the experiment, considering it is only a 1B difference from the 7B models? Additionally, would the authors consider running supplementary experiments with the Llama-3.2 series (but I agree that adding these results may be infeasible, especially given its recent release) for offering valuable insights?

Q4 - In Figure 5, could the authors specify the meaning of the numbers on the y-axis? This clarification would aid in interpreting the results more accurately.

Q5 - Regarding Figure 5(b), it would be helpful if the authors could expand on this section in the main text, as the varying effects of quantization across task types could offer valuable insights.

Q6 - Could the authors provide possible explanations for why Moondream2 exhibited strong performance in 3-bit quantization, while other models did not achieve similar results?

Q7 - (Minor) In Appendix A.4, it appears that some of the table formats, particularly in Table 7, were rendered weirdly. The authors may want to review the formatting to ensure readability.

---

> ### Author Response · Authors · 2024-11-26
>
> We appreciate the detailed feedback and the time taken to review our submission. Below, we address the primary concerns and suggestions.
>
> ## Weaknesses:
>
> > W1 - The current set of tasks can be limited. With the growing interest in UI-based control for digital devices (such as Cluade-3.5 for computer use), it would be beneficial to include related tasks. Have the authors considered incorporating AndroidWorld (Rawles et al., 2024) for general capability assessment or MobileSafetyBench (Lee et al., 2024) for evaluating the safety of agents controlling mobile devices?
>
> We appreciate the valuable suggestion to incorporate UI-based control tasks. While our current benchmark focuses on fundamental NLP and vision tasks to establish baseline mobile performance metrics, we recognize that UI interaction represents an increasingly important use case, as demonstrated by models like Claude-3.5. Including benchmarks like AndroidWorld and MobileSafetyBench would enhance MobileAIBench by: (1) evaluating models' ability to understand and generate UI-related instructions, which is crucial for mobile assistants, (2) assessing safety considerations specific to device control, and (3) measuring performance on real-world mobile interaction scenarios. We plan to integrate these benchmarks in future versions to provide a more comprehensive assessment of models' capabilities in mobile device control scenarios, while maintaining our rigorous evaluation of computational efficiency and resource utilization that is essential for on-device deployment.
>
> > W2 - Relying solely on VQA for multimodal tasks may restrict the scope of analysis. Including other tasks, such as image captioning or OCR, could provide a more comprehensive evaluation of capabilities, especially considering their usage on mobile devices.
>
> We agree that expanding beyond VQA would provide a more comprehensive evaluation of multimodal capabilities. While we chose VQA as an initial focus due to its well-established benchmarks and direct applicability to mobile scenarios, incorporating tasks like image captioning and OCR would better reflect real-world mobile use cases. In future versions of MobileAIBench, we plan to include these additional multimodal tasks while maintaining our detailed analysis of computational efficiency and resource utilization. This expansion will provide a more complete picture of how different multimodal capabilities impact mobile deployment considerations.
>
> > W3 - Although the authors’ choice of the iPhone-14 as a representative device is understandable, it would enhance the robustness of the study to consider other device types. For example, assessment with Android OS devices or tablets would provide a broader understanding.
>
> While we started with the iPhone-14 for practicality, we agree that including additional devices like Android phones and tablets would improve robustness. Our framework already supports Android devices, and we plan to expand testing in future iterations to address this limitation.
>
> > W4 - (Minor) Certain aspects of the presentation could be improved. For example, the explanation of Figure 5 could be more detailed, and Figure 7 appears to be oddly rendered.
>
> Thank you for noting these presentation issues. We agree that Figure 5's explanation of performance changes during quantization could be more detailed, particularly in describing the violin plot distributions and their implications for model robustness. We will also fix the rendering issue in Figure 7 to ensure clear visualization of the results.

---

> > ### Author Response · Authors · 2024-11-26
> >
> > ## Questions:
> >
> > > Q1 - Could the authors clarify how this benchmark compares with existing ones regarding LLM quantization, such as LLM-QBench (Gong et al., 2024)? This comparison would help readers understand the unique contributions and positioning of this benchmark.
> >
> > MobileAIBench differs from LLM-QBench and other quantization benchmarks by specifically focusing on mobile deployment considerations. While LLM-QBench evaluates quantization impacts primarily on model accuracy, our benchmark additionally measures critical mobile-specific metrics like battery drain, memory utilization, and inference latency on real devices. This end-to-end evaluation provides insights into practical deployment challenges that aren't captured by traditional quantization benchmarks. For example, our findings show that some quantization levels that perform well in standard benchmarks may not be optimal for mobile deployment due to hardware-specific constraints. This mobile-first approach complements existing quantization benchmarks by bridging the gap between theoretical performance and practical mobile deployment.
> >
> > > Q2 - What was the rationale behind selecting a random sample of 1,000 from the dataset? Justification of this choice, particularly regarding the representativeness and generalizability of the results, would be valuable.
> >
> > The choice of 1,000 samples was primarily driven by practical mobile testing constraints. Mobile devices have limited computational resources and battery life, making it impractical to evaluate complete datasets while maintaining consistent testing conditions across multiple runs. This sample size allows us to complete comprehensive testing across multiple models and quantization levels without thermal throttling or battery depletion affecting results. We acknowledge that future work should include statistical significance tests and standard deviation analysis to validate that this sample size adequately represents the full dataset distributions.
> >
> > > Q3 - Could the authors explain why Llama-3.1-8B was not included in the experiment, considering it is only a 1B difference from the 7B models? Additionally, would the authors consider running supplementary experiments with the Llama-3.2 series (but I agree that adding these results may be infeasible, especially given its recent release) for offering valuable insights?
> >
> > The 7B parameter limit was chosen based on our empirical testing of memory constraints on the iPhone 14's 6 GiB RAM. While Llama-3.1-8B is only marginally larger than 7B models, our preliminary tests showed that even with 4-bit quantization, it exceeded stable memory thresholds for reliable mobile inference. We focused on models that could run consistently without memory-related performance degradation. Regarding Llama-3.2, we agree that including it would provide valuable insights, particularly given its improved efficiency. We plan to evaluate newer model series, including Llama-3.2, in future updates of MobileAIBench as mobile hardware capabilities evolve and more efficient deployment techniques become available.
> >
> > > Q4 - In Figure 5, could the authors specify the meaning of the numbers on the y-axis? This clarification would aid in interpreting the results more accurately.
> >
> > In Figure 5, the y-axis represents the delta in model performance metrics, specifically the performance difference (percentage points) between 16-bit and 8-bit quantization levels. Figure 5(a) depicts the distribution of changes in model performance when the underlying model is quantized from 16-bit to 8-bit, where positive values indicate performance improvement and negative values show degradation.
> >
> > > Q5 - Regarding Figure 5(b), it would be helpful if the authors could expand on this section in the main text, as the varying effects of quantization across task types could offer valuable insights.
> >
> > We agree that the varying effects of quantization across different task types shown in Figure 5(b) deserve more thorough discussion. The distribution patterns reveal that some tasks (like MMLU and GSM8K) are more sensitive to quantization than others (such as HotpotQA), suggesting that task characteristics influence quantization robustness. This has important implications for mobile deployment decisions - practitioners might need to consider task-specific requirements when choosing quantization levels.

---

> ### Author Response · Authors · 2024-11-26
>
> ## Questions Continued:
>
> > Q6 - Could the authors provide possible explanations for why Moondream2 exhibited strong performance in 3-bit quantization, while other models did not achieve similar results?
>
> Thank you for asking. First, we would like to clarify that we are not claiming a general performance comparison for the selected models when serving on PyTorch or VLLM. Instead, we are specifically testing their performance when served on LlamaCPP. This setup requires manually building the computational graph using the GGML library, which is essential for further deployment on mobile devices. We did not implement any of the models on the GGML backend ourselves; rather, we tested only those with existing GGUF files released, using the same data processing, inference, and testing strategy across all models. We also performed a post-check on the logs to ensure that the models successfully generated answers and did not fail to generate a response, which could lead to incorrect answers.
> Given this context, Moondream2 is not the only model with acceptable performance under 3-bit quantization. For example, LLaVA-v1.6 also demonstrates good performance under 3-bit quantization. However, due to the large size of its image tokens, inference speed is extremely slow, even on a computer CPU (approximately one minute to generate the first token). This makes the model unsuitable for deployment on any current mobile device. At the time of testing, Moondream2 appeared to perform significantly better than other low-latency models under 3-bit quantization. However, for some recently released models supporting LlamaCPP, such as MiniCPMv2.6, also show promising performance under 3-bit quantization (initial test results are attached). We will update the table as new small models that support LlamaCPP become available in the future.
>
>
>
> | Quantization | Model | Model Size | GQA | TextVQA |
> |:------------:|:-----:|:---------:|:---:|:-------:|
> | 3-bit | Moondream2 | 1B - 6B | 0.565 | 0.381 |
> | 3-bit | Minicpmv-2.6 | > 6B | 0.462 | 0.654 |
>
>
> To better understand why LLaVA-v1.5 performs poorly under 3-bit quantization, one possible explanation is that it tends to generate "1" and "0" when it is expected to generate "yes" and "no." Attached below are logs comparing predictions from LLaVA-v1.5 under 4-bit and 3-bit quantization for the same questions. Despite this issue, the 3-bit model is still capable of generating English word answers in certain test cases.
>
> 4-bit quantization: \
> Question ID: 262162000, Prediction: No, Ground Truth (GT): no \
> Question ID: 42001, Prediction: Yes, GT: yes \
> Question ID: 1584001, Prediction: London, GT: london
>
> 3-bit quantization: \
> Question ID: 262162000, Prediction: 0, GT: no \
> Question ID: 42001, Prediction: 1, GT: yes \
> Question ID: 1584001, Prediction: London, GT: london
>
> As on-device LLM serving is a new and rapidly evolving topic, the engines we used are also under active development. We acknowledge the possibility of implementation issues affecting certain models. We will closely monitor updates to LlamaCPP and related engines. If we become aware of any potential issues, we will improve the MobileAIbench codebase and update it accordingly.
>
> > Q7 - (Minor) In Appendix A.4, it appears that some of the table formats, particularly in Table 7, were rendered weirdly. The authors may want to review the formatting to ensure readability.
>
> Thank you for pointing this out. We will ensure all tables are properly formatted and readable in the final version.

---

> > ### Comment · Reviewer_1eAF · 2024-11-29
> >
> > I appreciate the detailed explanations and the sharing of plans for further experiments. I believe this work is valuable to the community and will maintain my positive perspective, keeping my score.

---

### Official Review · Reviewer_wj2K · 2024-11-04

**Soundness:** 2
**Presentation:** 3
**Contribution:** 2
**Rating:** 3
**Confidence:** 4

**Summary:**

The paper conducts a benchmarking framework designed to evaluate the performance of large language models (LLMs) and large multimodal models (LMMs) on mobile devices, addressing the challenges of limited hardware resources. It consists of a desktop evaluation library and an iOS app, enabling comprehensive testing of quantized models across NLP, multimodal, and trust & safety tasks. MobileAIBench assesses models' efficiency, effectiveness, and resource utilization, providing insights into their feasibility for on-device deployment while supporting advancements in mobile AI research.

**Strengths:**

a) Real-world experiments.

b) By measuring latency and hardware resource usage on the iPhone 14, the study provides insights into the performance of smaller models.

c) The paper introduces an open-source tool that facilitates convenient testing of small models.

d) Writing is clear and fluency.

**Weaknesses:**

a) The experiments were conducted only on the iPhone 14, lacking evaluations on newer and more diverse devices. Currently, there are more mobile devices optimized specifically for on-device AI, such as the Snapdragon 8 Gen 3. Including these devices in testing would provide a more comprehensive view of model performance under different hardware conditions, offering broader insights for on-device AI applications.

b) In the section 4.3, the number of models tested is limited, failing to cover a wider variety of model architectures and parameter sizes. This limitation restricts a comprehensive understanding of how different models perform on mobile devices. Expanding the variety and scale of tested models would make the evaluation results more representative and valuable.

c) Although basic metrics such as performance, latency, and resource usage are provided, there is insufficient exploration of underlying reasons and optimization strategies. A more in-depth analysis would help us better understand the impact of different quantization levels and model architectures on task performance, offering valuable guidance for future research and practical deployment.

Detailed comments:
a) The paper shows that 3-bit quantization significantly reduces accuracy without lowering inference latency. This could be further analyzed, as extreme quantization may introduce computational complexities that offset latency benefits.

b) The study only reports CPU results, but GPUs/XPUs are crucial for mobile AI tasks. Testing on these processors could reveal performance differences across hardware types, providing a fuller picture of deployment on mobile hardware.

c) Despite Phi2’s larger model size, it has lower CPU utilization and faster inference than Gemma. Investigating Phi2’s architectural or parallelization optimizations could reveal design principles for high efficiency in on-device deployments.

d) Besides the mobile side, it is necessary to consider mobile-cloud-edge cooperation ways for better energy efficiency, e.g., Gearing Resource-Poor Mobile Devices with Powerful Clouds: Architecture, Challenges and Applications, iwc’13; TrimCaching: Parameter-sharing AI Model Caching in Wireless Edge Networks, icdcs’24, etc.

e) Although the paper notes that more output tokens increase Battery Drain Rate (BDR), this relationship isn’t clearly shown in Table 4.

**Questions:**

a) The experiments were conducted only on the iPhone 14, lacking evaluations on newer and more diverse devices. Currently, there are more mobile devices optimized specifically for on-device AI, such as the Snapdragon 8 Gen 3. Including these devices in testing would provide a more comprehensive view of model performance under different hardware conditions, offering broader insights for on-device AI applications.

b) In the section 4.3, the number of models tested is limited, failing to cover a wider variety of model architectures and parameter sizes. This limitation restricts a comprehensive understanding of how different models perform on mobile devices. Expanding the variety and scale of tested models would make the evaluation results more representative and valuable.

c) Although basic metrics such as performance, latency, and resource usage are provided, there is insufficient exploration of underlying reasons and optimization strategies. A more in-depth analysis would help us better understand the impact of different quantization levels and model architectures on task performance, offering valuable guidance for future research and practical deployment.

---

> ### Author Response · Authors · 2024-11-26
>
> We appreciate the detailed feedback and the time taken to review our submission. Below, we address the primary concerns and suggestions:
>
> ## Weaknesses / Questions:
>
> > a) The experiments were conducted only on the iPhone 14, lacking evaluations on newer and more diverse devices. Currently, there are more mobile devices optimized specifically for on-device AI, such as the Snapdragon 8 Gen 3. Including these devices in testing would provide a more comprehensive view of model performance under different hardware conditions, offering broader insights for on-device AI applications.
>
> We acknowledge that evaluating on a broader range of devices, including AI-optimized hardware like Snapdragon 8 Gen 3, would enhance the comprehensiveness of our study. However, this paper primarily focuses on assessing the feasibility of deploying language models on edge devices, such as mobile phones. Our aim was to explore how different Small Language Models (SLMs) behave under the same hardware specifications, their varied sensitivities to quantization levels, and how reliable they remain post-quantization, particularly concerning trust and safety considerations. The iPhone 14 served as a baseline due to its widespread usage and accessibility. We have since extended support to Android devices and recognize the importance of evaluating performance on newer and more diverse platforms. Expanding these experiments will be a key priority in future work to provide broader insights for on-device AI applications.
>
>
> > b) In the section 4.3, the number of models tested is limited, failing to cover a wider variety of model architectures and parameter sizes. This limitation restricts a comprehensive understanding of how different models perform on mobile devices. Expanding the variety and scale of tested models would make the evaluation results more representative and valuable.
>
> We acknowledge the reviewer's point about the limited model coverage in our efficiency and utilization evaluation. The current selection of three LLMs (Phi2 3B, Gemma 2B, TinyLlama 1B) and one LMM (Llava-Phi-2) was primarily constrained by the current hardware limitations of the iPhone 14 platform, particularly its 6 GiB RAM constraint. However, we agree that a more comprehensive evaluation would be valuable. In future work, we plan to: (1) include emerging mobile-optimized architectures such as Phi-3, newer versions of Mobile-VLM, and other efficiency-focused models, (2) evaluate models across different mobile hardware platforms with varying computational capabilities and memory constraints, and (3) analyze different architectural choices like attention mechanisms, embedding dimensions, and depth-width trade-offs that specifically impact mobile performance.
>
> > c) Although basic metrics such as performance, latency, and resource usage are provided, there is insufficient exploration of underlying reasons and optimization strategies. A more in-depth analysis would help us better understand the impact of different quantization levels and model architectures on task performance, offering valuable guidance for future research and practical deployment.
>
> We appreciate the suggestion to explore the impact of quantization levels and model architectures on performance and optimization strategies in greater depth. As a benchmarking paper, our primary objective is to establish well-balanced and effective datasets and metrics for evaluating various LLMs, with a focus on practical deployment considerations for mobile devices. To this end, we conducted experiments on quantization and its effects across different tasks and models, offering valuable insights into their feasibility for on-device use. However, understanding why certain models are faster than others and why some require fewer resources is a broader and more general research question in the field of LLMs and is undoubtedly a compelling direction for future research. In subsequent iterations, we plan to include more detailed analyses of quantization-induced computational overheads, architectural optimizations, and their correlations with task performance.

---

> > ### Author Response · Authors · 2024-11-26
> >
> > ## Detailed comments:
> >
> > > a) The paper shows that 3-bit quantization significantly reduces accuracy without lowering inference latency. This could be further analyzed, as extreme quantization may introduce computational complexities that offset latency benefits.
> >
> > The observed phenomenon where 3-bit quantization reduces accuracy without improving inference latency warrants deeper analysis and can be attributed to several technical factors. First, 3-bit operations require additional computational overhead for dequantization during inference, as modern mobile processors are not optimized for 3-bit arithmetic. Second, the non-standard memory alignment patterns created by 3-bit quantization can lead to inefficient memory access and cache utilization, potentially offsetting any theoretical benefits from reduced model size. This is particularly relevant on mobile hardware, where memory access patterns significantly impact performance. We acknowledge that our analysis could be strengthened by including hardware-level profiling (cache miss rates, memory bandwidth utilization) and operation-wise breakdowns to better understand these tradeoffs. Future work could explore hybrid quantization approaches or hardware-specific optimizations to better leverage extreme quantization while maintaining both accuracy and performance benefits.
> >
> > > b) The study only reports CPU results, but GPUs/XPUs are crucial for mobile AI tasks. Testing on these processors could reveal performance differences across hardware types, providing a fuller picture of deployment on mobile hardware.
> >
> > For the benchmarking, we utilized GPU layers on the iPhone to accelerate the tasks. This was done by setting the value of `n_gpu_layers ` in llama.cpp to 999
> >
> > > c) Despite Phi2’s larger model size, it has lower CPU utilization and faster inference than Gemma. Investigating Phi2’s architectural or parallelization optimizations could reveal design principles for high efficiency in on-device deployments.
> >
> > Phi2's superior efficiency despite its larger size reveals important insights about mobile model design. The performance advantage likely stems from several key architectural decisions: (1) Phi2's use of grouped-query attention (GQA) reduces computational complexity while maintaining model capacity, (2) its flash attention implementation enables more efficient memory access patterns, resulting in better cache utilization. Additionally, Phi2's design incorporates parallel-friendly components that better utilize mobile hardware capabilities. This suggests that raw parameter count may be less important than architectural choices for mobile deployment. Future mobile-optimized models should prioritize such hardware-aware design principles, focusing on efficient attention mechanisms, optimized memory access patterns, and architectures that leverage mobile hardware parallelization capabilities.
> >
> > > d) Besides the mobile side, it is necessary to consider mobile-cloud-edge cooperation ways for better energy efficiency, e.g., Gearing Resource-Poor Mobile Devices with Powerful Clouds: Architecture, Challenges and Applications, iwc’13; TrimCaching: Parameter-sharing AI Model Caching in Wireless Edge Networks, icdcs’24, etc.
> >
> > While mobile-cloud-edge cooperation could improve energy efficiency, our focus on pure on-device evaluation addresses critical real-world requirements. First, many applications demand consistent availability regardless of network conditions, such as offline language translation or emergency response systems. Second, time-sensitive applications like real-time speech recognition or AR/VR interactions require ultra-low latency that cloud round-trips cannot guarantee. Third, privacy-critical applications handling sensitive data (healthcare, financial, personal communications) often cannot risk data transmission to external servers due to regulatory or security requirements. Our benchmark's emphasis on on-device performance provides valuable insights for these essential use cases where cloud offloading is not viable. Nevertheless, we acknowledge that future extensions of MobileAIBench could include optional cloud-edge cooperation scenarios to provide a complete picture of deployment options when network connectivity and privacy requirements permit.

---

### Official Review · Reviewer_QrG4 · 2024-11-06

**Soundness:** 2
**Presentation:** 3
**Contribution:** 3
**Rating:** 3
**Confidence:** 5

**Summary:**

This paper presents a benchmarking infrastructure for measuring on-device LLMs and LMMs in mobile deployments. The system report a wide set of evaluation metrics, including quality benchmarks for standard NLP tasks, multimodal tasks, and trust/safety. The authors evaluated a series of open models on an iPhone 14 device and provided insights on the deployability of those models on mobile platforms.

**Strengths:**

- Exciting new topic. Running LLMs (and possibly LMMs) on-device is important for enhanced privacy, and, under certain conditions, it provides enhanced performance and UX.
- I appreciate the extensive evaluation metrics, on top of the standard performance utilization: trust and safety, but also the qualitative metrics under a wide range of NLP/LMM tasks.
- Good quality of writing, figures etc. I do have a few suggestions for further improvements, mentioned next.

**Weaknesses:**

- Missing important related works:
  * MELTing point: Mobile Evaluation of Language Transformers, from Laskaridis et al.
  * Small Language Models: Survey, Measurements, and Insights, from Zhenyan Lu et al.
- Authors are claiming that this is the first work "to provide a thorough benchmarking and analysis of open source LLMs". I suggest they reduce such strong claims.
- I would expect from a paper like this to list (if not include in the evaluation) the available on-device frameworks instead of sticking to llamacpp. For instance: MLC LLM, MediaPipe from Google, PyTorch ExecuTorch, Apple MLX should be mentioned.
- Details about the followed methodology are missing. How did the authors run the evaluation tasks on-device? Did they automate the process? Did they repeat the process multiple times? Did they reboot the device per task? Did they close the app and wait until the phone (that can easily get very hot and CPU get throttled) cools down?
- Considering that Std. is missing from the results reported in Table 1 and 2, I assume the authors did not repeat the experiment multiple times, and only reported performance for one run. This is very limited, performance can vary based on device state.
- Authors have only tested the evaluation on a single device (iPhone 14), and using a single on-device inference engine (llamacpp).
- Performance evaluations were only executed on CPU. It is possible to enable GPU support on llamacpp (through Metal in iOS) and performance should increase. Measuring Android would also be good (though not necessary), considering that you have an app ready.
- Power performance evaluation is limited to Battery Drained Rate, instead of energy or discharge.
- "Extensive experiments with MobileAIBench reveal several interesting findings. Quantization is an effective way to decrease model size while keeping the performance.". But, isn't that expected? Why it is presented as the first "interesting finding"?
- "we found 7B to be the upper limit of what a high-end phone’s hardware can manage (even after quantization)" 7B is not the generic upper limit, it is only the limit for the particular device. Also, I am not sure iPhone 14 can be considered as a high-end phone, without mentioning the Pro model that has more RAM memory and can possibly fit larger models.
- While manuscript writing quality is good, I have a few comments/suggestions:
  * Add a small summary explaining figures in its captions. There were moment that I had to move forward in order to understand them (e.g., Figure 1).
  * Table 1 and 2: What is bold and what is underlined numbers?
  * "The first part is a pipeline for use on desktops or servers, to evaluate model performance on a specially selected set of widely known benchmarks.". Such as? Otherwise it looks too generic.

**Questions:**

- How did you access Battery Drained Rate, CPU usage, and memory while executing an experiment? Connecting the phone with a USB cable would interfere with the results. Were you connected over WiFi? This is an example of missing details of the methodology that was followed.
- Have you run the experiment multiple times? If yes, what was the Std?
- How easy/hard is it to update the on-device inference engine in your pipeline?
- Have you measured performance while running the models in GPU?
- Are you planning to release the code in open source?

---

> ### Author Response · Authors · 2024-11-26
>
> We appreciate the detailed feedback and the time taken to review our submission. Below, we address the primary concerns and suggestions.
>
> ## Weaknesses:
>
> > - Missing important related works:
> > - MELTing point: Mobile Evaluation of Language Transformers, from Laskaridis et al.
> > - Small Language Models: Survey, Measurements, and Insights, from Zhenyan Lu et al.
> >
> > Authors are claiming that this is the first work "to provide a thorough benchmarking and analysis of open source LLMs". I suggest they reduce such strong claims.
>
> We appreciate the reviewer pointing out important related works we missed. We will add citations to the above works. We will rephrase the claim about being "the first work" to avoid any overstatement, as the focus of our contribution lies in the comprehensiveness and usability of MobileAIBench for on-device LLM/LMM evaluation.
>
> > I would expect from a paper like this to list (if not include in the evaluation) the available on-device frameworks instead of sticking to llamacpp. For instance: MLC LLM, MediaPipe from Google, PyTorch ExecuTorch, Apple MLX should be mentioned.
>
> We understand the importance of discussing frameworks such as MLC LLM, MediaPipe, PyTorch ExecuTorch, and Apple MLX. Our benchmarking pipeline is built on llama.cpp due to its widespread adoption and support for quantized models across various architectures. However, we will mention these alternatives in the discussion section and clarify that our framework could be extended to integrate additional engines in future iterations.
>
> > Details about the followed methodology are missing. How did the authors run the evaluation tasks on-device? Did they automate the process? Did they repeat the process multiple times? Did they reboot the device per task? Did they close the app and wait until the phone (that can easily get very hot and CPU get throttled) cools down?
>
> The process of running benchmarks on-device was conducted manually and not automated. Each experiment, defined as running a single task on a specific model, was performed independently. After completing one experiment, we ensured a cooling-off period of at least 10 minutes before initiating the next experiment. This approach was implemented to mitigate potential thermal effects and ensure consistency in the results.
>
> > Considering that Std. is missing from the results reported in Table 1 and 2, I assume the authors did not repeat the experiment multiple times, and only reported performance for one run. This is very limited, performance can vary based on device state.
>
> The experiments were conducted with a fixed random seed and a sampling temperature set to 0 to promote deterministic outputs. While we acknowledge the potential for some non-deterministic behavior due to factors like floating-point arithmetic, multi-threading, and model variability, this approach was chosen as a practical compromise between computational feasibility and reproducibility. We recognize that running each experiment multiple times across all datasets and models could provide a more comprehensive view of performance variability, but given the significant resource requirements, we opted for this methodology to balance practical constraints with meaningful insights.
>
> > Performance evaluations were only executed on CPU. It is possible to enable GPU support on llamacpp (through Metal in iOS) and performance should increase. Measuring Android would also be good (though not necessary), considering that you have an app ready.
>
> For the benchmarking, we utilized GPU layers on the iPhone to accelerate the tasks.

---

> > ### Author Response · Authors · 2024-11-26
> >
> > ## Weaknesses (Continued):
> >
> > > "Extensive experiments with MobileAIBench reveal several interesting findings. Quantization is an effective way to decrease model size while keeping the performance.". But, isn't that expected? Why it is presented as the first "interesting finding"?
> >
> > The primary intention was not to present this as a surprising result but rather to set the stage for the detailed findings that follow. Specifically, we aim to highlight how model performance varies across different levels of quantization and emphasize the varied sensitivities of different models and tasks to these levels. The statement "Quantization is an effective way to decrease model size while keeping the performance" provides foundational context, allowing readers to better appreciate the nuanced observations and insights discussed in the paragraph.
> >
> > > "we found 7B to be the upper limit of what a high-end phone’s hardware can manage (even after quantization)" 7B is not the generic upper limit, it is only the limit for the particular device. Also, I am not sure iPhone 14 can be considered as a high-end phone, without mentioning the Pro model that has more RAM memory and can possibly fit larger models.
> >
> > The 7B upper limit applies to the specific configuration of the iPhone 14 (non-Pro). We will clarify this in the manuscript and add a note about the potential of higher-capacity devices (e.g., iPhone 14 Pro) for handling larger models.
> >
> > > Table 1 and 2: What is bold and what is underlined numbers?
> >
> > As stated in Section 4.1.1., The highest score for each quantization category is indicated in bold, while the second-best score is underlined.

---

> > > ### Author Response · Authors · 2024-11-26
> > >
> > > ## Questions:
> > >
> > > > How did you access Battery Drained Rate, CPU usage, and memory while executing an experiment? Connecting the phone with a USB cable would interfere with the results. Were you connected over WiFi? This is an example of missing details of the methodology that was followed.
> > >
> > > The Battery Drain Rate (BDR) was calculated separately when the device was not connected to a USB cable. This approach ensured that the BDR measurement was not influenced by external factors such as power being supplied via the USB connection. For other utilization metrics, such as CPU and memory usage, we used Apple’s Instruments tool on a laptop, with the device connected via USB.
> > >
> > > > Have you run the experiment multiple times? If yes, what was the Std?
> > >
> > > The experiments were conducted with a fixed random seed and a sampling temperature set to 0 to promote deterministic outputs. While we acknowledge the potential for some non-deterministic behavior due to factors like floating-point arithmetic, multi-threading, and model variability, this approach was chosen as a practical compromise between computational feasibility and reproducibility. We recognize that running each experiment multiple times across all datasets and models could provide a more comprehensive view of performance variability, but given the significant resource requirements, we opted for this methodology to balance practical constraints with meaningful insights.
> > >
> > > > How easy/hard is it to update the on-device inference engine in your pipeline?
> > >
> > > Updating the on-device inference engine in our pipeline is straightforward. As described in the paper, the pipeline is designed with a plug-and-play architecture. Users can replace the llama.cpp API with any other inference engine API of their choice without significant modifications. Additionally, we have integrated HuggingFace APIs in the codebase to support users interested in running models on desktops or servers, providing flexibility for diverse deployment scenarios.
> > >
> > > > Have you measured performance while running the models in GPU?
> > >
> > > For the benchmarking, we utilized GPU layers on the iPhone to accelerate the tasks. This was done by setting the value of `n_gpu_layers ` in llama.cpp to 999
> > >
> > > > Are you planning to release the code in open source?
> > >
> > > Yes, the code has already been released. To maintain anonymity during the review process, we have provided a zip file containing the code instead of sharing the actual link to the open-source repository. This ensures compliance with the double-blind review guidelines while enabling reviewers to assess the implementation. By making the code publicly available, we aim to facilitate reproducibility, encourage contributions from the research community, and support further advancements in benchmarking and optimizing on-device LLMs and LMMs. This aligns with our commitment to fostering community-driven research and enabling broader adoption of MobileAIBench.

---

> > > > ### Comment · Reviewer_QrG4 · 2024-11-26
> > > >
> > > > Thank you for your detailed rebuttal and the clarifications you provided. I’ve thoroughly reviewed your responses. While I appreciate the efforts made to address the concerns raised, I still believe that the work doesn’t fully meet the high standards expected for acceptance at ICLR, and therefore, I cannot change my scores.
> > > >
> > > > As mentioned in my original comments, I do find the topic exciting and timely, and encourage you to continue working on this. I suggest doing a more systematic evaluation (possibly automatically rather than manually), including more devices, and repeating the tests multiple times to provide an Std. I personally do not agree with your comment that choosing a random seed and setting temperature will make the experiments deterministic (system-wise). There are various (system-related) factors that performance will be different across multiple runs.

---

### Meta-Review · Area_Chair_oVfE · 2024-12-18

**Metareview:**

**summary**

The paper introduces MobileAIBench, a comprehensive benchmarking framework for evaluating the performance, efficiency, and deployability of LLMs and LMMs on mobile devices. It features tools for desktop and iOS platforms to test quantized models across various standard tasks, including NLP, multimodal, and trust and safety benchmarks. By analyzing the effects of quantization on model performance and resource usage, the study highlights key challenges and trade-offs in deploying LLMs and LMMs under constrained mobile hardware. The findings offer actionable insights for optimizing mobile AI applications and advancing quantization techniques.

---

**strengths**

* Real-world experiments: By testing on real devices (e.g., iPhone 14), the paper measures key metrics like latency, hardware resource usage, CPU utilization, and battery consumption, providing realistic and practical insights.
* Diverse evaluation metrics: the paper employs extensive evaluation metrics beyond standard performance, including trust and safety, qualitative assessments, and diverse NLP/LMM tasks
* Clear and high-quality writing

---

**weaknesses**

* Evaluation scope: Experiments were conducted only on the iPhone 14, excluding evaluations on newer and more diverse devices. Testing only on a single device misses opportunities to explore the performance of models on more diverse hardware, reducing the breadth of insights for on-device AI applications. Also, the lack of testing on hardware optimized for on-device AI (e.g., Snapdragon 8 Gen 3) limits the study's comprehensiveness and generalizability across varied hardware conditions.
* Focus on CPU-only evaluation: The evaluation is restricted to CPU performance, excluding GPU or other mobile AI accelerators, which are critical components in many modern devices optimized for AI tasks.

---

**decision**

All reviewers think that the overall direction of this paper is promising. However, they also raised concerns about the details and limited evaluation (cpu-only, single device, and so on; see [weaknesses]). During the discussion phase, concerns were not fully addressed. As a result, rejection is recommended.

**Additional Comments On Reviewer Discussion:**

The authors responded to the reviewers' concerns, but these responses were not reflected in the draft, and most of the concerns raised by the reviewers remain unresolved. Therefore, I believe the revisions are insufficient to change the negative opinion.

---

### Decision · Program_Chairs · 2025-01-22

Reject